# Exonuclease action of replicative polymerase gamma drives damage-induced mitochondrial DNA clearance

Akshaya Seshadri [iD][1,2] & Anjana Badrinarayanan [iD][1✉]

## Abstract

**Mitochondrial DNA (mtDNA) replication is essential for mitochondrial function. This is carried out by a dedicated DNA polymerase gamma, with 5'–3' polymerase and 3'–5' proofreading/exonuclease activity. Perturbations to either property can have pathological consequences. Predominant sources for replication stress are DNA lesions, such as those induced by oxidative damage. How mtDNA lesions affect the polymerase activity and mtDNA stability in vivo is not fully understood. To address this, we induce mtDNA-specific damage in *S. cerevisiae*. We observe that mtDNA damage results in significant mtDNA loss. This loss occurs independent of cell cycle progression or cell division, suggesting an active mechanism for damaged mtDNA clearance. We implicate the 3'–5' exonuclease activity of the mtDNA polymerase in this clearance, with rates of loss being affected by cellular dNTP levels. Overall, our findings reveal context-dependent, selective regulation of two critical but opposing functions of polymerase gamma to ensure mitochondrial genome integrity.**

**Keywords** mtDNA Damage; PolG; Mip1; DNA Replication; Proofreading
**Subject Categories** DNA Replication, Recombination & Repair; Organelles

## Introduction

Faithful replication and maintenance of mitochondrial DNA (mtDNA) is essential for mitochondrial function and cell survival under respiratory growth conditions. Most eukaryotic cells contain multiple copies of their mtDNA, packaged into distinct mtDNA nucleoids (Chen and Butow, 2005). The number of nucleoids increases as a function of cell volume (Seel et al, 2023). Replication of mtDNA within these nucleoids occurs independently of nuclear genome replication (Bogenhagen and Clayton, 1977), (Sasaki et al, 2017), and is carried out by a dedicated DNA polymerase, Polymerase gamma (PolG) (Genga et al, 1986), (Lestienne, 1987), (Copeland and Longley, 2014). PolG belongs to the A-family of polymerases, which includes *E. coli*

DnaI and the T7 DNA polymerase (Kaguni, 2004). In metazoans, PolG is a trimeric holoenzyme: a PolGα subunit with 5'–3' polymerase and 3'–5' exonuclease domains, and a dimeric PolGβ that increases the processivity of PolGα, via regulation of both polymerase and exonuclease activity (Oliveira et al, 2015). The budding yeast ortholog of PolG (Mip1) is a processive enzyme encoding the 5'–3' polymerase and 3'–5' exonuclease domains in a single polypeptide (Viikov et al, 2011). In vitro, PolG (without exonuclease activity) displays an error rate of $15 \times 10^{-6}$ per nucleotide (Longley et al, 2001). The synthesis-associated error rate is reduced to $<8 \times 10^{-6}$ per nucleotide by the exonuclease domain, that recognizes mis-paired 3' termini during synthesis (Foury and Vanderstraeten, 1992a), (Kaguni and Olson, 1989). In vivo, mutations in exonuclease function can increase mtDNA mutation rates by several 100-fold (Foury et al, 2004) and mice models of exonuclease-deficient PolG have displayed elevated mtDNA mutations, cardiomyopathy, and premature ageing (Copeland and Longley, 2014).

In addition to maintaining replication fidelity at the site of DNA synthesis, the exonuclease domain of the mitochondrial polymerase plays important roles in other contexts as well. Given the polyploid nature of mtDNA, mutant DNA molecules typically constitute only a small fraction of the total mtDNA. It has been suggested that elimination of the damaged mtDNA copies, combined with the replication of the intact ones could be a mechanism by which mtDNA integrity and function is maintained (Shokolenko and Alexeyev, 2015). In several instances, the exonuclease domain has been implicated in such clearance. For example, linear mtDNA fragments (generated by action of restriction endonucleases) are subject to degradation by the exonuclease domain of PolG (Nissanka et al, 2018), (Peeva et al, 2018). Furthermore, in conditions of extreme starvation in yeast, it has been shown that the exonuclease domain of Mip1 results in mtDNA clearance, likely as a salvage pathway to replenish dNTP levels (Medeiros et al, 2018). Interestingly, in Trypanosoma, one of the six mtDNA polymerases was recently shown to function as an exonuclease only, as it lacked the catalytic activities associated with polymerase function (Delzell et al, 2022).

It remains unclear as to how the exonucleolytic function, that is typically restricted to only a few bases around the terminal nucleotides in a primer strand, is extended to these different capacities, while still maintaining a balance between the polymerase

---

[1]National Centre for Biological Sciences - Tata Institute of Fundamental Research, Bangalore, Karnataka, India. [2]School of Chemical and Biotechnology, SASTRA University, Thanjavur, Tamil Nadu, India. ✉E-mail: anjana@ncbs.res.in

and exonuclease activity. Indeed, imbalance in the same can have direct consequences on replication fidelity as well as total amount of DNA synthesis carried out by the polymerase (Szczepanowska and Foury, 2010). Even within the context of mtDNA replication, the polymerase is likely to encounter replication-stalling lesions (such as those generated by oxidative damage) frequently during DNA synthesis, in part due to the reactive oxygen species-rich environment of the mitochondria (Anderson et al, 2020). This makes mtDNA susceptible to damage significantly more than its nuclear counterpart (Yakes and Van Houten, 1997), with point mutation rates of mtDNA reported to be ~10-fold higher than the nuclear genome (Parsons et al, 1997), (Haag-Liautard et al, 2008). How the polymerase responds to such DNA lesions in vivo, and how polymerase/exonuclease balance is maintained to ensure mtDNA integrity in such a scenario remains unknown.

We previously developed a tool to induce mtDNA-specific damage using a mitochondrially targeted bacterial toxin, DarT (mtDarT) (Dua et al, 2022). This toxin generates adducts on single-stranded DNA, creating replication-stalling lesions specifically on mtDNA (Dua et al, 2022; Jankevicius et al, 2016; Lawarée et al, 2020). In fermentative conditions, where yeast cells can proliferate without mtDNA or its associated functions, such mtDNA damage resulted in the reorganization of the mitochondrial network and asymmetric segregation of mtDNA-less mitochondria, with an acceleration in this process observed in cells carrying a mutation in the exonuclease domain of PolG (Mip1 henceforth) (Dua et al, 2022). In this study, we utilized mtDarT to understand how such lesions affect mtDNA maintenance and Mip1 function in vivo in *S. cerevisiae*, in conditions where oxidative phosphorylation (and hence mitochondrial function) is essential for cell proliferation. We specifically monitored the impact of mtDNA lesions on mtDNA copy number. We found that mtDNA damage resulted in mtDNA loss. This loss was independent of cell cycle progression or cell division, suggesting an active mechanism of mtDNA clearance. In support, we uncovered a role for the $3'$–$5'$ exonuclease of Mip1 in this mtDNA loss under damage, with cellular dNTP levels regulating the rates of clearance. We propose a model where Mip1 encounter with a DNA lesion could switch its function from synthesis to degradation, thus resulting in the clearance of the lesion-containing mtDNA. Together, our work highlights the selective regulation of polymerase/exonuclease balance for the maintenance of mtDNA integrity under persistent DNA damage.

## Results

### mtDNA damage results in mtDNA loss

To study the effect of mtDNA damage on mtDNA maintenance, we utilized a tool that was previously characterized by our group to specifically damage mtDNA in *S. cerevisiae* (Dua et al, 2022). For this, we expressed mitochondrially targeted DarT (mtDarT) from a β-estradiol-inducible promoter in yeast cells grown in non-fermentable glycerol-containing medium (YPG), with cells maintained in mid-exponential growth throughout the course of the experiment (Dua et al, 2022) (Fig. EV1A). We first confirmed the expression of mtDarT in YPG medium via Western blotting and found robust induction of mtDarT 4 and 8 h after the addition of β-estradiol (Fig. EV1B). Furthermore, cells expressing mtDarT were

compromised in survival, as assessed by their growth on YPG plates carrying β-estradiol. This was in contrast to empty vector control cells, which did not show a growth defect on β-estradiol containing YPG plates (Fig. EV1C). We then proceeded to monitor the fate of mtDNA in cells subjected to damage, via SYBR Green I (SGI) staining of mtDNA nucleoids in live cells (Jajoo et al, 2016). As a control, we compared mtDarT-expressing cells with cells carrying an empty vector and subjected to the same experimental regime (Fig. EV1A).

In the absence of damage, pulse-staining with SGI resulted in the labeling of mtDNA nucleoids (Jajoo et al, 2016), that were seen as distinct cytoplasmic puncta in the cell (Fig. 1A). These puncta were localized within the mitochondria (Fig. EV1D), and no nuclear DNA staining was observed (Fig. EV1D). As a control, SGI staining of $\rho^0$ cells (which lack mtDNA) resulted in nuclear DNA staining and no cytoplasmic puncta were observed (Fig. EV1E, and as reported previously (Baruffini et al, 2010)), suggesting that the cytoplasmic puncta we were visualizing were indeed mitochondrial nucleoids. In the absence of damage, multiple mtDNA foci were observed in all cells (empty vector, Fig. 1A). We found that the number of nucleoids per cell increased as a function of cell area, in line with previous observations showing mtDNA nucleoid scaling with cell size (Seel et al, 2023) (empty vector (0 h), Spearman's correlation coefficient ($\rho$) 0.76, CI: 12.43503, 13.18894) (Fig. 1B). The percentage of cells with mtDNA foci as well as the relationship of the number of nucleoids with cell area was maintained in empty vector cells, even after 8 h of β-estradiol treatment (Fig. 1B,C).

In cells carrying the mtDarT expression vector, we observed a modest increase in the percentage of cells without mtDNA foci even without damage induction, likely due to leaky expression of mtDarT (mtDarT (0hr) (Fig. 1A–C). This increase was significantly pronounced upon induction of mtDarT, with fewer nucleoids per cell observed when compared to the empty vector control (Fig. 1A,B, mtDarT). At 8 h post induction, the relationship between number of nucleoids and cell area was significantly perturbed (Spearman's correlation coefficient ($\rho$) 0.53, CI: 8.851388, 9.779604) (Fig. 1B), and the percentage of cells without mtDNA foci increased gradually, with 32±4% cells without mtDNA foci observed after 8 h of mtDarT induction (Fig. 1C). Indeed, majority of cells had fewer than 10 nucleoids per cell 8 h after damage induction (Fig. 1D). Consistent with the depletion of mtDNA in these conditions, we found that mtDarT-expressing cells frequently displayed nuclear DNA staining as well (Fig. 1A). Together, these observations suggest that mtDNA damage results in mtDNA loss under non-fermentative growth conditions.

### mtDNA loss occurs independent of cell cycle progression

What cellular mechanisms might result in the observed loss of mtDNA? We first considered the possibility that asymmetric partitioning of damaged mtDNA during division could result in the generation of $\rho^0$ cells as we had previously seen in fermentative growth conditions (Dua and Badrinarayanan, 2023). In this scenario, cells at each division would receive fewer numbers of nucleoids (or no nucleoids at all). To test this possibility, we arrested cells in either G1/S or G2/M stage of the cell cycle and then induced mtDarT to follow the fate of mtDNA under damage in non-dividing cells. For this, we first pre-treated Δ*bar1* cells (*MAT*a) with α-factor which resulted in a G1/S phase arrest (Fig. 2A), following which we induced mtDNA damage in

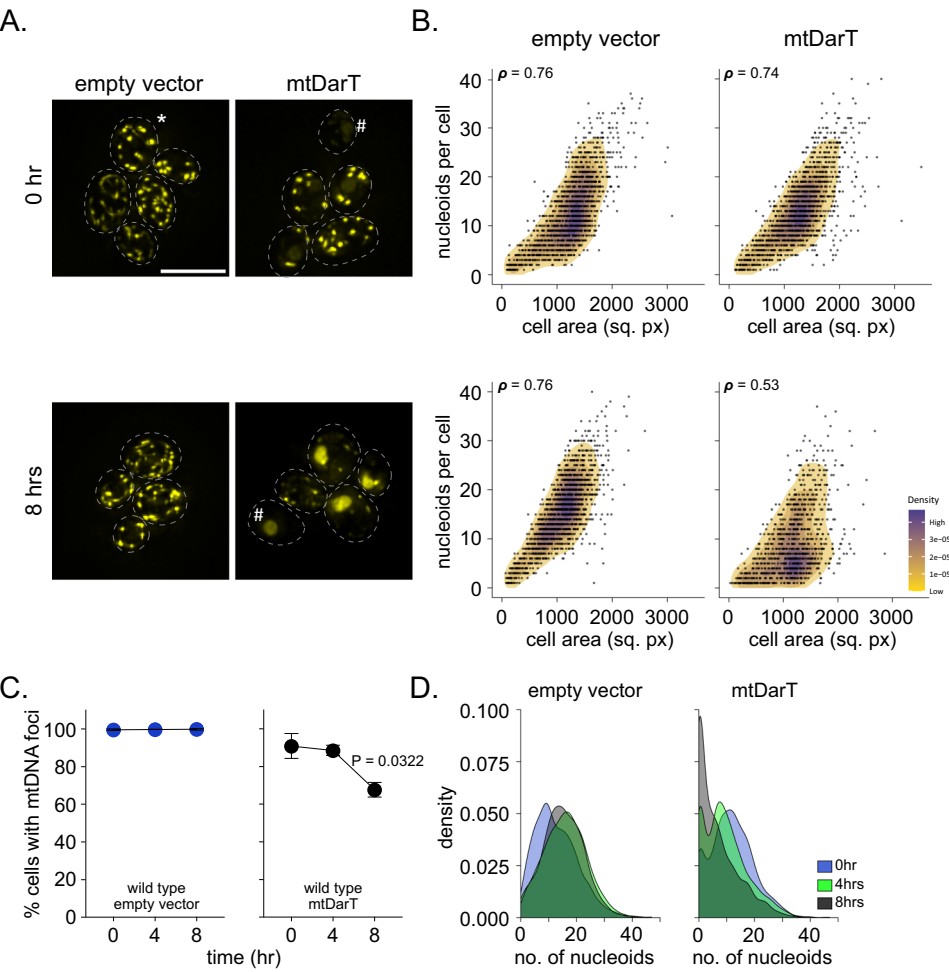

**Figure 1. mtDNA damage results in mtDNA clearance.**

(A) Representative images of mitochondrial nucleoids stained with Sybr Green I (SGI). All experiments were performed in YPG medium. Top: nucleoids in empty vector and mtDarT cells before addition of β-estradiol (0 h). Bottom: nucleoids in empty vector and mtDarT cells after the addition of β-estradiol (8 h). Dashed lines represent cell boundaries. *Indicates mt-nucleoids and # indicates cells devoid of mtDNA staining. (B) 2D density correlation plots of the number of nucleoids per cell against the respective cell area for empty vector or mtDarT before (top) and 8 h post induction (bottom). Spearman's correlation coefficient (ρ) is indicated in the top-left corner of each plots. Data are pooled from three independent repeats (n >287 cells per group, per repeat). (C) Percentage of cells containing mtDNA foci in empty vector (left) or mtDarT (right) expressing cells at 0, 4, and 8 h after induction. Data are shown from three independent repeats (n >207 cells per group, per repeat). Mean and SD are shown. Significance was calculated using repeated measures one-way ANOVA and post hoc tests. *P ≤0.05. (D) Density distributions for the number of nucleoids per cell in empty vector (left) and mtDarT (right) expressing cells before (0 h) and after (4, 8 h) induction. Data are pooled from three independent repeats (n >273 cells per group, per repeat). Scale bar, 8 μm here, and in all other images. Statistical tests and exact P values for all graphs are provided in Dataset EV1. Source data are available online for this figure.

this arrested population. As a control, damage was also induced in an asynchronously growing cell culture for comparison. We observed that the number of nucleoids was reduced in arrested cells after 8 h of damage, with few to no nucleoids in these highly elongated cells (Fig. 2B). In addition, the percentage of cells with mtDNA foci was comparable between asynchronous (72.4%) and arrested (70.5%) populations after 8 h of damage (Fig. 2C). Similarly, the relationship between cell size and the number of nucleoids was also perturbed in the arrested cells expressing mtDarT ($\rho = 0.44$ (CI: 12.65272, 15.20398)) (Fig. 2D). In contrast, empty vector cells had densely packed nucleoid foci dispersed throughout the cell (Fig. 2B) and the relationship between cell size and number of nucleoids per cell was also maintained ($\rho = 0.87$ (CI: 33.81381, 40.35107)) (Fig. 2D). Thus, G1-/S-arrested cells showed mtDNA loss under damage in respiratory growth conditions

(Fig. 2C–E). To eliminate any potential influence from the cell cycle stage under arrest, we separately investigated mtDNA loss in nocodazole-induced G2/M-arrested cells (Fig. EV2A). Upon mtDarT expression, here too we observed a decrease in the percentage of cells with mtDNA foci (Fig. EV2B), similar to that observed in G1/S-arrested cells. Taken together, these observations indicate that cell cycle progression (and cell division) does not contribute to mtDNA loss under mtDNA damage.

## mtDNA loss is dependent on Mip1 exonuclease

An important source for mtDNA loss could be mitophagy, which can drive the clearance of dysfunctional mitochondria (with mutant mtDNA). Such dysfunction-induced mitochondrial clearance has

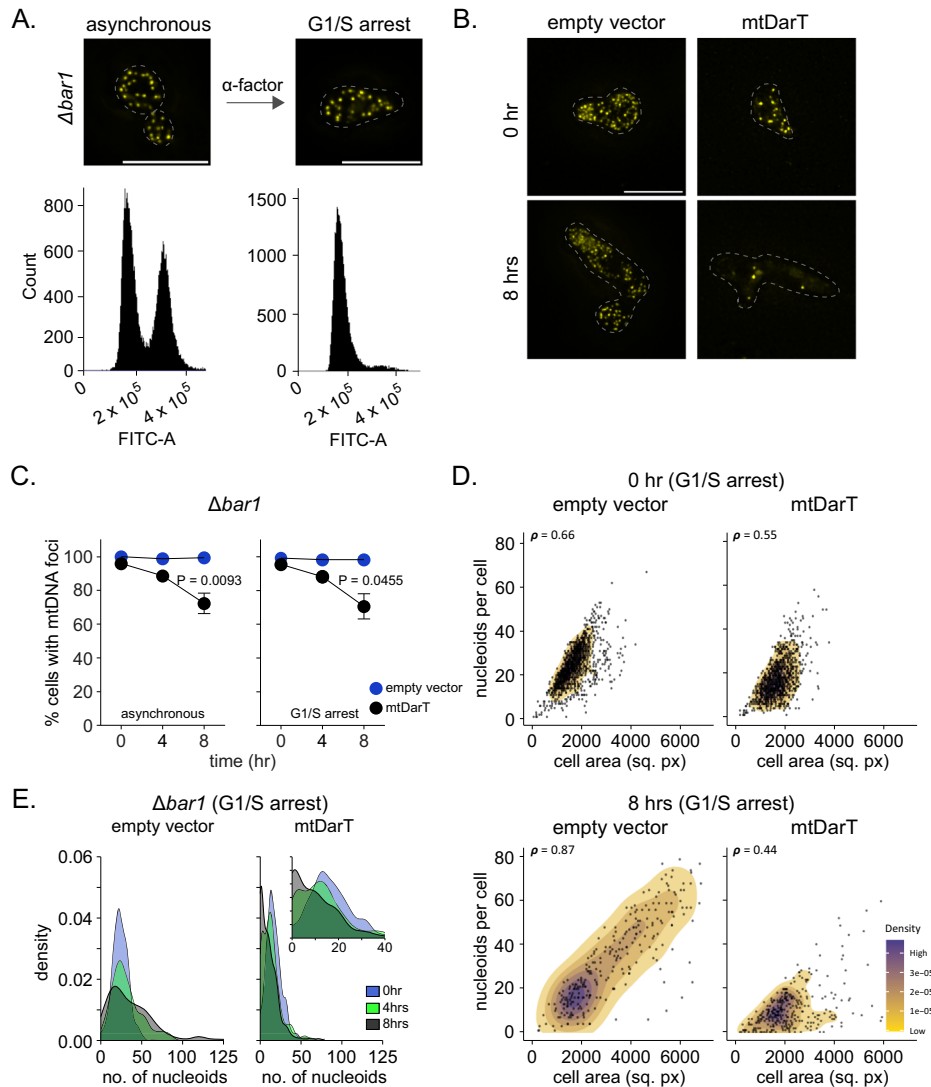

**Figure 2. mtDNA loss occurs independent of cell cycle progression.**

(A) Experimental setup followed for inducing mtDNA damage under cell cycle arrest using α-factor in Δbar1 cells. Representative images of nucleoids in asynchronous and arrested cells are shown. Flow cytometry profiles of cell cycle stages in asynchronous and arrested cells are also provided. Dashed lines represent cell boundaries. (B) Representative images of nucleoid foci in G1-/S-arrested cells. Top: empty vector and mtDarT-expressing cells at 0 h. Bottom: empty vector and mtDarT-expressing cells at 8 h post induction. Dashed lines represent cell boundaries. (C) Percentage of cells with mtDNA foci in empty vector (blue) or mtDarT (black) expressing cells at 0, 4, and 8 h after induction in (left) asynchronous population and (right) in G1/S-arrested cells. Data shown from three independent repeats (n >61 cells per group, per repeat). Mean and SD are shown. Significance was calculated using repeated measures one-way ANOVA and post hoc tests. *P ≤ 0.05 and **P ≤ 0.01. (D) 2D density correlation plots of the number of nucleoids per cell against the respective cell area in G1/S-arrested cells for empty vector or mtDarT before (top) and 8 h post induction (bottom). Spearman's correlation coefficient (ρ) is indicated in the top-left corner of each plots. Data are pooled from four independent repeats (n >50 cells per group, per repeat). (E) Density distributions for the number of nucleoids per cell in empty vector (left) or mtDarT (right) expressing cells before (0 h) and after (4, 8 h) induction under G1/S arrest. Data are pooled from four independent repeats (n > 50 cells per group, per repeat). Inset in mtDarT shows distribution up to 40 nucleoids/cell. Scale bar, 8 μm here, and in all other images. Source data are available online for this figure.

been reported in case of *Drosophila* cells carrying a mutation on mtDNA (Kandul et al, 2016). To assess if this occurs in the case of mtDNA damage in budding yeast, we deleted the receptor essential for mitophagy in budding yeast (Atg32) (Kanki et al, 2009). We observed that mitophagy was not required for mtDNA loss under damage, with a non-significant difference in the percentage of cells with mtDNA nucleoids between wild-type and Atg32-deleted cells even after 8 h of damage induction (Fig. EV3A). Furthermore, we did not observe a significant change in Cox4 protein levels under

damage conditions, suggesting that mitochondrial content did not significantly change upon damage induction (Fig. EV3B). These observations are also consistent with our previous findings that mitophagy (which is typically observed in much longer timescales) may not play a role under mtDNA damage in yeast (Dua et al, 2022).

Since mtDNA loss in response to mtDNA damage occurred even in non-dividing cells, we considered the possibility of an active mtDNA clearance mechanism, such as a nuclease-mediated DNA

degradation under damage. In *S. cerevisiae*, Nuc1, homolog to the mammalian EndoG, is the major mitochondrial endonuclease (Dzierzbicki et al, 2012). In order to check if Nuc1 was responsible for clearing out mtDNA under damage, we deleted Nuc1 and tested the effect of the same on mtDNA nucleoid numbers. We observed that Nuc1 was not essential for mtDNA loss under damage (Fig. EV3C).

Another nuclease that has been connected with mtDNA clearance is the exonuclease domain of the mitochondrial replicative polymerase (PolG/ Mip1). In mammalian cells, PolG exonuclease domain contributes to the degradation of linear mtDNA fragments (Nissanka et al, 2018), (Peeva et al, 2018). Similarly, in yeast cells facing extreme starvation, the exonuclease domain of Mip1 has been implicated in mtDNA clearance (Medeiros et al, 2018). Whether such activity is also relevant in the context of mtDNA damage is unknown. We thus assessed whether the exonuclease activity of Mip1 could also result in mtDNA degradation under conditions of mtDNA damage.

Based on previously characterized mutations, we generated an exonuclease-deficient Mip1 via mutating 2 catalytic residues in the ExoI motif of the enzyme, Aspartate 171 (Asp171) to Alanine 171 (Ala171) and Glutamate 173 (Glu173) to Alanine 173 (Ala173) (*mip1*$^{exo-}$ henceforth) (Foury and Vanderstraeten, 1992b), (Medeiros et al, 2018), (Longley et al, 1998) (Fig. 3A). Both Mip1 and *mip1*$^{exo-}$ were expressed from their endogenous locus on the chromosome. Western blot experiments confirmed that their expression in mtDarT-induced samples were comparable (Fig. EV3D). We did note that *mip1*$^{exo-}$ cells expressing mtDarT showed an elevated percentage of cells without mtDNA foci prior to damage induction (Fig. 3C). However, upon mtDarT induction, *mip1*$^{exo-}$ cells maintained nucleoids even after 8 h of damage, and damage-induced mtDNA depletion was not observed as was seen in case of wild-type cells (Fig. 3B–E). Although there was no loss of nucleoids, the relationship between number of nucleoids and cell area was still perturbed in this background (Spearman's correlation coefficient ($\rho$) 0.44, CI: 13.54795, 14.56954) (Fig. 3D, mtDarT). These observations suggest that the exonuclease activity of the mitochondrial replicative polymerase drives mtDNA loss under DNA damage. The compromised scaling of nucleoid numbers with increasing cell size even in the absence of exonuclease activity of Mip1 suggests that replication progression could be additionally compromised under mtDNA damage.

## Exonuclease activity of Mip1 drives mtDNA loss under damage

To test whether exonuclease activity was directly driving mtDNA clearance under damage, we complemented the *mip1*$^{exo-}$ strain with ectopic expression of the exonuclease domain in this background. For this, we generated a variant of *mip1* (Mip1-CΔ263) expressed from the *MIP1* promoter on a low copy replicating plasmid where the last 263 amino acids in the C-terminus of Mip1 are deleted. This mutant has been extensively characterized previously, and in vitro studies have shown that it is deficient in polymerization activity, but still retains the ability of polymerase to bind DNA, as well as an intact exonuclease domain and function (Trasviña-Arenas et al, 2019; Young et al, 2021). We confirmed the expression of the truncated protein via western blot and that the protein colocalized with mitochondrial nucleoids (Fig. EV4A,B). Similar to

wild-type cells (Fig. 4A), we observed that expression of Mip1-CΔ263 in *mip1*$^{exo-}$ background resulted in a significant decrease in the percentage of cells containing mtDNA foci upon mtDNA damage induction (Fig. 4B). So as to confirm that exonuclease activity from the Mip1-CΔ263 construct was indeed contributing to the observed mtDNA loss, we introduced the exonuclease mutation in the truncation construct. Western blot of both Mip1-CΔ263 and the exonuclease-deficient Mip1-CΔ263 (*mip1*$^{exo-}$-CΔ263) showed similar expression levels, and this protein also colocalized with mtDNA (Fig. EV4A,B). Furthermore, under damage, we no longer observed mtDNA loss occurring in the absence of exonuclease activity from this exonuclease-deficient construct when expressed in *mip1*$^{exo-}$ strain (Fig. 4C). Based on these results, we conclude that Mip1-CΔ263 does indeed harbor exonuclease activity that drives the observed mtDNA loss under damage.

To further assess the role of the exonuclease domain of Mip1 in mtDNA clearance, we generated two additional mutants in highly conserved regions of the polymerase domain that should be compromised in polymerase activity, while still retaining exonuclease function. These variants (Mip1$^{G651S}$) and (Mip1$^{H734Y}$) are located in the thumb and finger sub-domains and are reported to compromise catalysis and selectivity in nucleotide binding, respectively (Kasiviswanathan et al, 2009). Both mutants are orthologous to PolG mutations, G848S and H932Y, falling in regions of high amino acid homology among several organisms including budding yeast (Fig. EV4D) (Stumpf et al, 2010). We found that both mutants were expressed in cells and colocalized with mtDNA (Fig. EV4C,E). These mutants were able to complement *mip1*$^{exo-}$, with mtDNA loss being observed upon damage induction (Fig. 4D,E). Together, these results support a direct role for the exonuclease domain of Mip1 in mtDNA clearance under conditions of mtDNA damage.

## Rates of mtDNA loss are influenced by dNTP levels

How does Mip1 exonuclease activity result in mtDNA clearance under damage? Given that the polymerase and exonuclease domains are both present on the same protein in the case of Mip1, we wondered whether exonuclease-mediated clearance occurred on mtDNA that is being actively replicated. Studies in *S. cerevisiae* have shown that elevation in dNTP levels can increase Mip1 polymerase function (Lecrenier and Foury, 1995; Medeiros et al, 2018; Vanderstraeten et al, 1998; Viikov et al, 2012). We thus hypothesized that increasing polymerase activity (and hence access) on mtDNA should impact the rates of mtDNA loss under damage. To assess this, we deleted Sml1, a negative regulator of ribonucleotide reductase (RNR) activity in *S. cerevisiae*, that results in an increase in cellular dNTP levels (Zhao et al, 1998), and assessed the rates of mtDNA clearance under damage in this background.

We confirmed that the expression of Mip1 in Δ*sml1* background was comparable to that in wild-type cells under damage (Fig. EV5A). Furthermore, Δ*sml1* cells grew similarly to wild-type cells in non-fermentable media (Fig. EV5B), and had comparable number of mtDNA nucleoids in the absence of damage (Fig. 5A). Upon mtDarT induction, Δ*sml1* cells showed a significant increase in mtDNA loss (Fig. 5B,C). In comparison to wild type, the rates of mtDNA loss were higher in Δ*sml1* cells. After 8 h of damage induction, the percentage of cells without mtDNA foci were found to be 41±6% and 29±1% in Δ*sml1* and wild-type cells respectively (Fig. 5B). Importantly, this elevated mtDNA loss in Δ*sml1* cells was

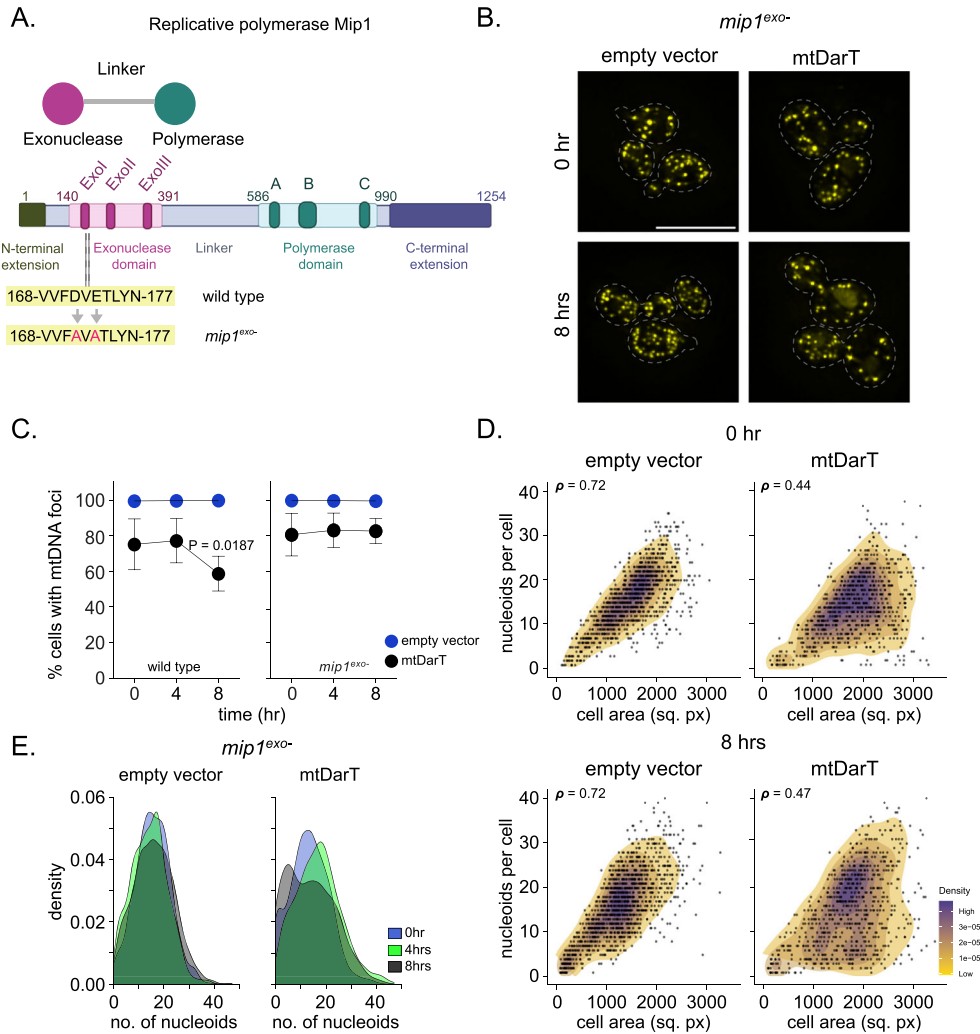

Figure 3. mtDNA loss is dependent on Mip1 exonuclease.

(A) Schematic representing the domain organization of Mip1. Residues mutated to generate *mip1^exo-* mutant are indicated. Schematic created with Biorendor.com. (B) Representative images of mt-nucleoid foci in *mip1^exo-* cells. Top: empty vector and mtDarT-expressing cells at 0 h. Bottom: empty vector and mtDarT-expressing cells at 8 h. Dashed lines represent cell boundaries. (C) Percentage of cells with mtDNA foci in empty vector (blue) or mtDarT (black) at 0, 4, and 8 h after induction for wild-type (left) and *mip1^exo-* cells (right). Data shown from four independent repeats (n >139 cells per group, per repeat). Mean and SD are shown. Significance was calculated using repeated measures one-way ANOVA and post hoc tests. *$P \leq 0.05$. (D) 2D density correlation plots of the number of mt-nucleoids per cell against the respective cell area for empty vector or mtDarT before (top) and 8 h post induction (bottom) in *mip1^exo-* cells. Spearman's correlation coefficient (ρ) is indicated in the top-left corner of each plots. Data are pooled from 3 independent repeats (n >149 cells per group, per repeat). (E) Density distributions for the number of nucleoids per cell before (0 h) and after (4, 8 h) induction in empty vector (left) and mtDarT (right) expressing *mip1^exo-* cells. Data are pooled from three independent repeats (n >88 cells per group, per repeat). Scale bar, 8 μm here, and in all other images. Source data are available online for this figure.

dependent on the exonuclease activity of Mip1, with no loss being observed in *Δsml1 mip1^exo-* cells (Fig. 5D).

In sum, our observations implicate the exonuclease domain of Mip1 in mtDNA clearance under conditions of mtDNA damage, with dNTP levels affecting the rates of this mtDNA loss.

## Discussion

Taken together, our study places the exonuclease activity of PolG as a central mediator of mtDNA loss under mtDNA damage conditions. Thus, the clearance of damaged mtDNA copies employs the very apparatus that replicates it. Such a mechanism

of damaged mtDNA clearance would restrict the degradative function of the polymerase to the damaged mtDNA alone, and circumvent the requirement to have an additional pathway to sense and respond to the damage more globally. In support, we did not observe a significant contribution from mitophagy in this process. While we cannot rule out the possibility that this damage clearance mechanism may contribute to mtDNA loss at much longer timescales, our observations are consistent with other reports suggesting that mitophagy may not play a significant role in mtDNA damage clearance in case of yeast cells (as opposed to metazoan systems) (Dua et al, 2022) (Abeliovich, 2023). Indeed, such drastic DNA loss might only be tolerated in a multicopy genome, allowing cells to lose most copies yet retain the capacity to

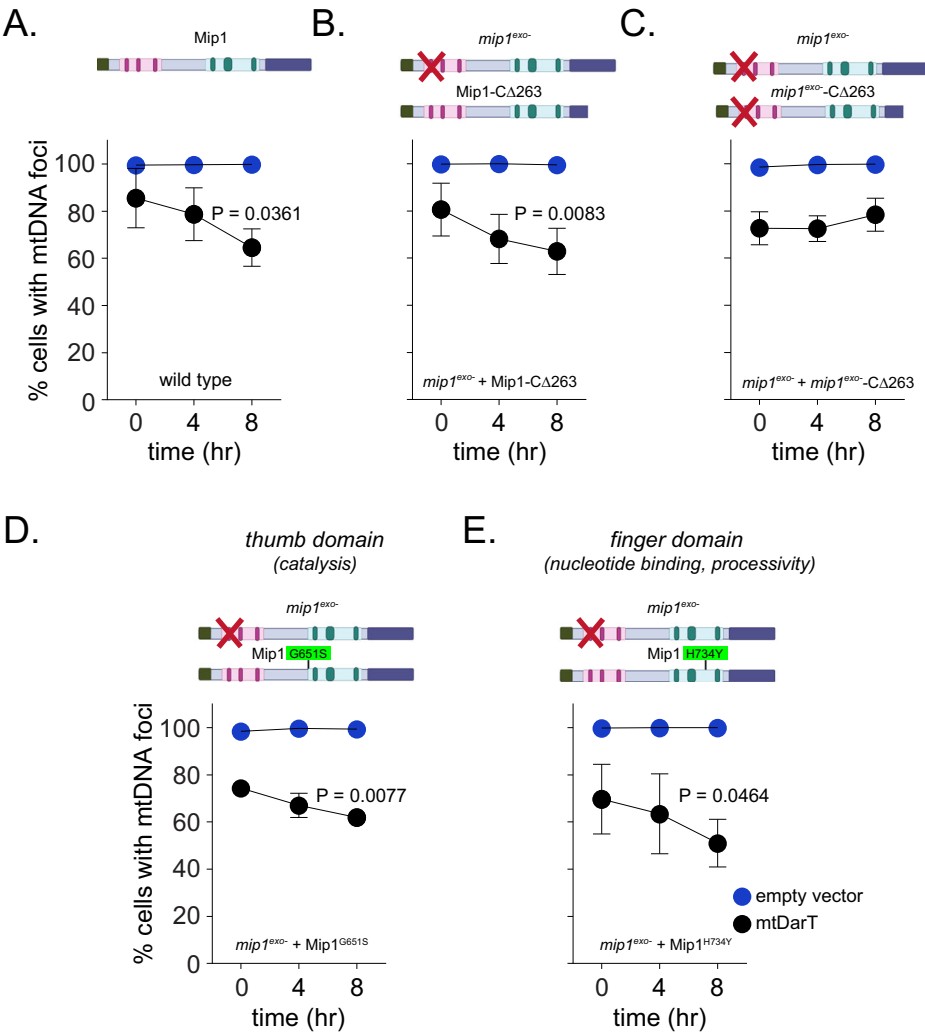

**Figure 4. Exonuclease activity of Mip1 drives mtDNA loss under damage.**

(A–C) Percentage of cells with mtDNA foci in empty vector (blue) or mtDarT (black) at 0, 4, and 8 h after induction for wild-type (A), *mip1^exo-* cells expressing Mip1-CΔ263 (B) and *mip1^exo-* cells expressing *mip1^exo-*-CΔ263 (C). Data are from three independent repeats (*n* >127 cells per group, per repeat). Mean and SD are shown. Significance was calculated using repeated measures one-way ANOVA and post hoc tests. *$P \leq 0.05$ and **$P \leq 0.01$. (D, E) Percentage of cells with mtDNA foci in empty vector (blue) or mtDarT (black) at 0, 4, and 8 h after induction for *mip1^exo-* cells expressing Mip1^G651S (D) and *mip1^exo-* cells expressing Mip1^H734Y (E). Data shown from three independent repeats (*n* >119 cells for Mip1^G651S, and *n* > 73 cells for Mip1^H734Y per group, per repeat). Mean and SD are shown. Significance was calculated using repeated measures one-way ANOVA and post hoc tests. *$P \leq 0.05$ and **$P \leq 0.01$. Source data are available online for this figure.

restore homeostasis post-damage removal, in line with the 'disposable genome' hypothesis for mitochondria (Shokolenko and Alexeyev, 2015).

The modest loss of mtDNA nucleoids from the leaky expression of mtDarT observed in the absence of Mip1 exonuclease function suggests that mtDNA replication may be additionally compromised. However, our data strongly support a major role for exonuclease action in the mtDNA loss observed upon damage induction. For example, in the *mip1^exo-* mutant where polymerase is still active and present, cells do not lose mtDNA upon mtDarT expression (Fig. 3C). This highlights that a possible perturbation of replication alone under damage does not immediately reflect as significant mtDNA loss, at least in the timescales in which we track mtDNA.

How are contrasting functions of Mip1 polymerase and exonuclease selectively regulated in this context-dependent manner (replication vs degradation)? We do not yet fully understand the mechanism by which substrates for Mip1 exonuclease activity (such as nicks or breaks) would be generated at sites of mtDarT damage. Sources for this can include events such as collisions at stalled replisomes, breaks due to the instability of exposed ssDNA, or additional damage triggered by oxidative stress. Based on our observations of accelerated mtDNA clearance under damage in Δ*sml1* cells, it is tempting to hypothesize that the hand-off from synthesis to degradation might occur at the site of replication stalling. Such an arrangement would ensure that the exonucleolytic action of Mip1 is regulated to prevent promiscuous mtDNA clearance in the absence of DNA damage. In the context of

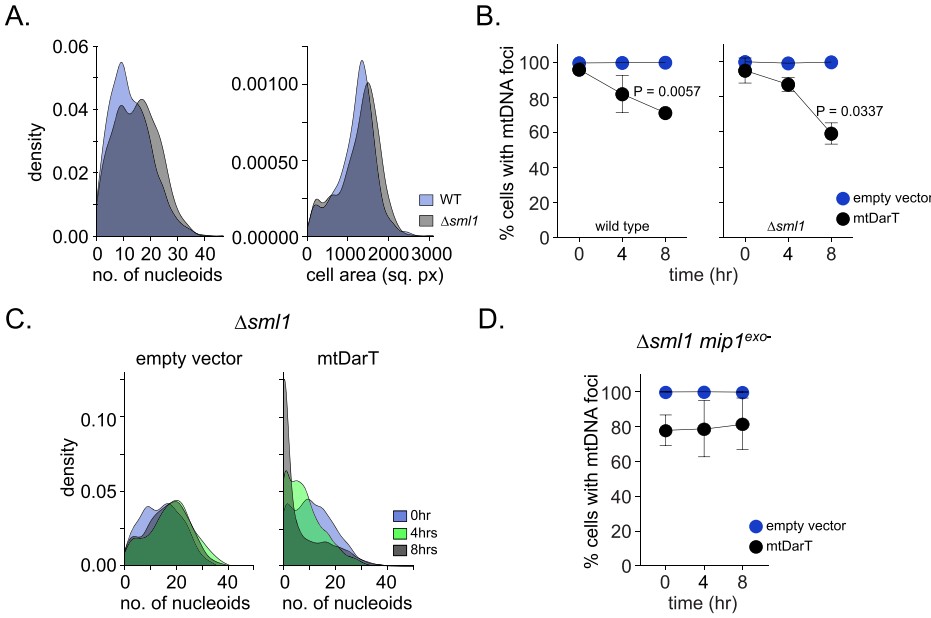

**Figure 5. Rates of mtDNA loss are influenced by dNTP levels.**

(A) Density distributions of the number of nucleoids per cell (left) and cell area (right) for wild-type (blue) and Δsml1 (gray) cells in the absence of damage. Data shown from three independent repeats (n >287 cells per group, per repeat). (B) Percentage of cells containing mtDNA foci in empty vector (blue) or mtDarT (black) at 0, 4, and 8 h after induction in wild-type (left) and Δsml1 (right) cells. Mean and SD are shown. Significance was calculated using repeated measures one-way ANOVA and post hoc tests. *P ≤ 0.05 and **P ≤ 0.01. Data shown from three independent repeats (n >203 cells per group, per repeat). (C) Density distributions for the number of nucleoids per cell in empty vector (left) or mtDarT (right) at 0, 4, and 8 h after induction in Δsml1 cells. (n >303 cells per group, per repeat). (D) Percentage of cells containing mtDNA foci in empty vector (left) or mtDarT (right) at 0, 4, and 8 h after induction in Δsml1 mip1exo- background. Data shown from three independent repeats (n >139 cells per group, per repeat). Mean and SD are shown. Significance was calculated using repeated measures one-way ANOVA and post hoc tests. Source data are available online for this figure.

heteroplasmy (Nissanka and Moraes, 2020), where only a fraction of mtDNA would typically carry a mutation, selective clearance would allow the retention of intact genomes over the mutant copies. In support, studies in *Drosophila* cells have shown preferential replication of non-mutated mtDNA copies (Hill et al, 2014). More recently, reports in budding yeast have also shown that mutant mtDNA are compartmentalized via modulation of the cristae morphology (Jakubke et al, 2021). It is possible that the exonuclease activity of Mip1 on these sequestered mtDNA would additionally contribute to the selective clearance of damaged genomes. How Mip1 accesses damaged mtDNA, how much DNA is degraded by a single exonuclease, and the dynamics of mtDNA resynthesis upon removal of the stress are exciting and outstanding questions.

Exonuclease activity of the replicative polymerase is canonically considered in the context of proofreading, where only few mismatched nucleotides from newly synthesized DNA are removed (Kunkel and Mosbaugh, 1989). However, more recent evidence suggest that the exonuclease domain of the mitochondrial replicative polymerase may play more versatile roles. Exonuclease action has previously been observed on linear mtDNA fragments (Nissanka et al, 2018), (Peeva et al, 2018) and under conditions of extreme starvation (where dNTP levels are depleted) (Medeiros et al, 2018). Exonuclease activity is also required for mtDNA ligation during replication (Macao et al, 2015), the absence of which results in the accumulation of mtDNA fragments. A previous

study from our group (Dua et al, 2022) and our current study both highlight the key role of Mip1 exonuclease in mtDNA loss under damage. In growth conditions with a fermentable carbon source (where yeast cells can continue to grow and divide even in the absence of functional mitochondria/ mtDNA), we had previously found that Mip1 exonuclease activity played a role in mtDNA clearance; in its absence, clearance rates were accelerated via asymmetric mtDNA segregation and cell division, resulting in rapid generation of ρ⁰ cells (Dua et al, 2022). When the possibility of viable cell divisions is compromised (such as in the present study using a non-fermentable carbon source, where mitochondrial function is essential for cell proliferation), Mip1 exonuclease appears to be the sole driver of this clearance during mtDNA stress.

The implication of exonuclease function in mtDNA clearance under conditions of mtDNA damage suggests that it is a central player in the maintenance of mtDNA integrity as well, beyond the regulation of replication-associated errors. Indeed, the choice to repair or degrade damaged mtDNA could be driven by the type and quantity of damage. For example, given the absence of active nucleotide excision repair in mitochondria (Alexeyev et al, 2013; Clayton et al, 1974), it is possible that the exonuclease-mediated pathway for mtDNA degradation may be the first response to persistent, replication-stalling lesions. Other contexts can include mtDNA deletions or mutations, where selective degradation of the damaged copies would be a reliable mechanism for ensuring mitochondrial genome integrity maintenance.

 

# Methods

## Reagents and tools table

| Reagent/resource | Reference or source | Identifier or catalog number |
| --- | --- | --- |
| **Experimental Models/Yeast strains** | | |
| CEN.PK (*S. cerevisiae*) Mat**a** | Van Dijken et al, 2000 | NABC567 |
| CEN.PK (*S. cerevisiae*) Matα | Van Dijken et al, 2000 | NABC568 |
| Δ*bar1* (*S. cerevisiae*) Mat**a** | Dr. Sunil Laxman Lab stock | NABC595 |
| Δ*sml1* (*S. cerevisiae*) Mat**a** | This study | NABC639 |
| Δ*nuc1* (*S. cerevisiae*) Mat**a** | This study | NABC641 |
| Mip1-mCherry (*S. cerevisiae*) Mat**a** | This study | NABC772 |
| Mip1-mNeongreen (*S. cerevisiae*) Mat**a** | This study | NABC773 |
| Mip1$^{exo-}$ (*S. cerevisiae*) Mat**a** | This study | NABC573 |
| Mip1$^{exo-}$mNeongreen (*S. cerevisiae*) Mat**a** | This study | NABC680 |
| Δ*sml1* Mip1-mNeongreen (*S. cerevisiae*) Mat**a** | This study | NABC766 |
| Cox4-mCherry (*S. cerevisiae*) Mat**a** | This study | NABC569 |
| Δ*atg32* (*S. cerevisiae*) Mat**a** | This study | NABC676 |
| **Recombinant DNA** | | |
| pFA6a-KO-Hyg | Dr. Sunil Laxman Lab stock | pNABC552 |
| pFA6a-KO-KanMX6 | Dr. Sunil Laxman Lab stock | pNABC551 |
| pFA6a-mCherry-Hyg | Dr. Sunil Laxman Lab stock | pNABC549 |
| pFA6a-mNeongreen-Kan | This study | pNABC553 |
| pGEV-3XFLAG-NAT | AB Lab stock, Dua et al, 2022 | NABC557 |
| pGEV-Su9-DarT-3XFlag-NAT | AB Lab stock, Dua et al, 2022 | pNABC561 |
| p417-cyc1-Kan | Dr. Sunil Laxman Lab stock | pNABC555 |
| pMip1-Mip1-CΔ263 in NABC555 background | This study | pNABC775 |
| pCYC1-Mip1-CΔ263-mNeongreen in NABC555 background | This study | pNABC777 |
| pMip1-Mip1-CΔ263-mCherry in NABC555 background | This study | pNABC943 |
| pMip1-mip1$^{exo-}$- CΔ263 in NABC555 background | This study | pNABC944 |
| pMip1-mip1$^{exo-}$- CΔ263-mCherry in NABC555 background | This study | pNABC945 |
| pMip1-mip1-G651S in NABC555 background | This study | pNABC946 |
| pMip1-mip1-G651S-mNeonGreen in NABC555 background | This study | pNABC947 |
| pMip1-mip1-H734Y in NABC555 background | This study | pNABC948 |
| pMip1-mip1-H734Y-mNeonGreen in NABC555 background | This study | pNABC949 |
| **Antibodies** | | |
| Rabbit Anti-Flag (1:500) | Cell Signalling | 14793S |
| Mouse Anti-mCherry (1:500) | Sigma | SAB2702291 |
| Rabbit Anti-mCherry (1:500) | Agrisera | AS18 4179 |
| Rabbit Anti-mNeongreen (1:500) | Agrisera | AS21 4525 |
| Anti-rabbit IgG, HRP-linked Antibody (1:10000) | Cell Signaling Technology | 7074S |
| Anti-mouse IgG, HRP-linked Antibody (1:10000) | Cell Signaling Technology | 7076S |
| **Oligonucleotides and other sequence-based reagents** | | |
| GGAGAACAGGAGAAAGAAGG | This study | AS78 |
| GGTTGAGCTGGAAAGGGACAT TACTATTTCTAGAGAGTACcggat ccccgggttaattaa | This study | AS79 |
| ATATAAATACAAATGCGAAA GCTAATGCAGATTTTGCCTAgaa ttcgagctcgtttaaac | This study | AS80 |
| TTAATTAACCCGGGGATCCG | This study | AS81 |
| CAAACAAGATCACCAAAAGG | This study | AS91 |
| AGTTCCCCACTTATTAGATCC CAAAGTTAAAGTTATCATGcgga tccccgggttaattaa | This study | AS92 |
| TTATGAATTAATTTATATTTACA GTTTTTCAGTACATTCAgaattcga gctcgtttaaac | This study | AS93 |
| ATGACGAAATTGATGGTTAGA TCTGAATGCATG | This study | AS112 |
| CTAGTACTCTCTAGAAATAGTA ATGTCCCTTTC | This study | AS113 |
| AATCGTCTTTCACAGTTTGC | This study | AS161 |
| ACAAAAGCAAAAAAAATCT GCCAGGAACAGTAAACATATG cggatccccgggttaattaa | This study | AS162 |
| AGTAGGAACGTGTATGTTT GTGTATATTGGAAAAAGGTT Agaattcgagctcgtttaaac | This study | AS163 |
| gctgccgccgctcgcgctgctgccgc tgccGCTGTCGATATCGAGAT TTGGTTTACC | This study | AS183 |
| ggcagcggcagcagcgcgagcggcg gcagcATGGCTTCTTTGCCAGC TACTC | This study | AS184 |
| CCCTCACTAAAGGGAACAA AAGCTGGAGCTCgcagaaagaac cttcaacctcattttgtt | This study | AS188 |
| GATATCGAATTCCTGCAGC CCGGGGGGATCCtcaagaattagaa gaagcagaacctgaacc | This study | AS192 |
| ggcagcggcagcagcgcgagcggcg gcagcATGGTGAGCAAGGGCGA | This study | AS200 |
| gatatcgaattcctgcagcccgggg gatccCTACTTGTACAGCTCG TCCATG | This study | AS201 |
| GATATCGAATTCCTGCAG CCCGGGGGATCCCTAgctgtc gatatcgagatttggtt | This study | AS202 |
| ggcagcggcagcagcgcgagcggc ggcagcATGGCTTCTTTGCCA GCTACTCATGAATTG | This study | AS225 |
| GAAAATCGTCCCCATGtc CACAATCACTAGAAGA | This study | AS271 |
| TCTTCTAGTGATTGTGgaCATGGGGACGATTTTC | This study | AS272 |
| GAAGGTACAGATTTGtACACGAAGACTGCTCAA | This study | AS273 |
| TTGAGCAGTCTTCGTGTaCAAATCTGTACCTTC | This study | AS274 |
| ACTAAACCCTGTTGGTGT TCCAAATGATGACCACCAT CACcggatccccgggttaattaa | This study | ND328 |
| GTAAAAGAGAAACAGAA GGGCAACTTGAATGATAAG ATTAgaattcgagctcgtttaaac | This study | ND329 |
| GGCATTAAATTATCGTCGTG | This study | ND307 |
| TCAAAAATTGATGGTGGAAG AAATAAATTTAGACATCGATcg gatccccgggttaattaa | This study | ND308 |
| ATTGATTTGAAAAGACCT CATATATTTACAAGAATATC TAgaattcgagctcgtttaaac | This study | ND309 |
| GAAAACTGCCCAAAACTTAG | This study | ND327 |
| GGGATTATATGTAGTTGTT GAGCAACGAGGGACAAGTAT Gcggatccccgggttaattaa | This study | ND334 |
| ATATAAATACAAATGCGAAAG CTAATGCAGATTTTGCCTAgaattc gagctcgtttaaac | This study | ND335 |

| Reagent/resource | Reference or source | Identifier or catalog number |
|---|---|---|
| TATGTAGTTGTTGAGCAACGA GGGACAAGTATGACGAAATTGA TGGTTAGATCTGAATGCATGCTG | This study | ND339 |
| CTGGTGGTGTTTGCTGTAGCA ACACTCTATAAC | This study | ND340 |
| CGTTATAGAGTGTTGCTACAG CAAACACCACCAG | This study | ND341 |
| CTAGTACTCTCTAGAAATAG TAATGTCCCTTTCCAG | This study | ND342 |
| AGGGACATTACTATTTCTAGA GAGTACTAGGACATGGAGGCCC AGAATACCCTCCTT | This study | ND343 |
| TACAAATGCGAAAGCTAATG CAGATTTTGCCTTCGAGCGTCC CAAAACCTTCTCAAGC | This study | ND344 |
| GGCCAATGATAGGAAAGAAC | This study | ND488 |
| CACTAACCTCTCTTCAACTGC TCAATAATTTCCCGCTATGcgga tccccgggttaattaa | This study | ND489 |
| AAATGGAAAGAGAAAAGAAA AGAGTATGAAAGGAACTTTAga attcgagctcgtttaaac | This study | ND490 |
| Chemicals, enzymes, and other reagents | | |
| SYBR™ Green I Nucleic Acid Gel Stain | Invitrogen | S7563 |
| Molecular Probes SYTOX Green Nucleic Acid Stain | Thermo Fisher Scientific | S7020 |
| AgriseraECL SuperBright | Agrisera | AS16 ECL-S |
| β-estradiol | Sigma-Aldrich | E2758 |
| Ultrapure agarose | Invitrogen | 16500500 |
| α1-Mating Factor acetate salt | Sigma-Aldrich | T6901 |
| Nocodazole | Sigma-Aldrich | 487928 |
| Hygromycin B | Thermo Fisher Scientific | 10687010 |
| Nourseothricin | Jena Bioscience | AB-102XL |
| DAPI (4',6-Diamidino-2-Phenylindole, Dihydrochloride) | Invitrogen | D1306 |
| Geneticin™ Selective Antibiotic (G418 Sulfate), Powder | Thermo Fisher Scientific | 11811031 |
| RNAse A | Genetix | EN0531 |
| Proteinase K | Invitrogen | AM2548 |
| Softwares | | |
| ImageJ FIJI 2.16.0 | | https://imagej.net/software/fiji/ |
| Graphpad Prism 8.4.2 | | https://www.graphpad.com |
| Cell ACDC | Padovani et al, 2022 | https://github.com/SchmollerLab/Cell_ACDC |
| RStudio Version 1.3.1093 | | https://posit.co/products/open-source/rstudio/ |
| NIS-elements software (version 5.01) | | Nikon Instruments Inc. |
| COBALT: multiple alignment tool | Papadopoulos and Agarwala, 2007 | https://www.ncbi.nlm.nih.gov/tools/cobalt/re_cobalt.cgi |
| Others | | |
| Accuri C6+ flow cytometer | | BD Biosciences |
| Nikon Eclipse Ti2-E | | Nikon Instruments Inc. |
| pE-4000 | | CoolLED |
| ORCA-Flash4.0 V3 Digital CMOS camera | | Hamamatsu Photonics |

## Yeast strains, media, and growth conditions

Strains, plasmids and oligos used in this study are listed in the Reagents and tools table. Cells were grown at 30 °C with 180 rpm shaking, in YP (yeast extract 1%, peptone 2%) medium. All experiments were performed in non-fermentable conditions using glycerol (2%) as the carbon source. Dextrose (2%, fermentable medium) was used for preparatory experiments, including transformations etc. All experiments were performed in

log phase of growth (OD600, between 0.2–0.6). Appropriate antibiotics and inducers were added to the medium as required. The strains generated were haploid prototrophs belonging to CEN.PK background (Van Dijken et al, 2000). For generating strains with C-terminal tagging or deletion of a genomic locus, PCR-based targeted homologous recombination strategy was used as described in (Longtine et al, 1998). Strains were verified by PCR, followed by fluorescence microscopy and/or western blotting for validating the insertion of epitope tags.

## Plasmid construction

### Plasmids expressing empty vector and mtDarT
Details of plasmids expressing the empty vector (pNABC557) and mtDarT (pNABC561) can be found in Dua et al, 2022.

### Mip1-CΔ263 (pNABC775)
The plasmid containing the Mip1-CΔ263 sequence (pNABC775), expressed under the pMip1 promoter, was generated by PCR amplification of the pMip1-Mip1-CΔ263 fragment (ending at Ser991 in Mip1) from wild-type yeast genomic DNA. The primers used were AS188 and AS202. A stop codon was introduced after residue 991. This fragment was then Gibson cloned into pNABC555 between SacI and BamHI sites, removing the cyc1 promoter.

### Mip1-CΔ263-mNeongreen (pNABC777)
A mNeongreen-tagged version of Mip1-CΔ263 (pNABC777) was created by Gibson cloning the Mip1-CΔ263 fragment (amplified using AS182 and AS163 from wild-type yeast genomic DNA) with a fragment containing a 10 amino acid linker followed by mNeongreen (amplified using AS184 and AS185). The combined fragment was then inserted into pNABC555 at the XbaI site.

### Mip1-CΔ263-mCherry (pNABC943)
To construct the mCherry-tagged Mip1-CΔ263 (pNABC943), the pMip1-Mip1-CΔ263 fragment (ending at Ser991 in Mip1) was amplified from wild-type yeast genomic DNA using primers AS188 and AS202. This fragment was then Gibson cloned with a fragment containing a 10 amino acid linker followed by mCherry (amplified using AS200 and AS201) into pNABC555 between SacI and BamHI sites, removing the cyc1 promoter.

Plasmids expressing $mip1^{exo-}$-CΔ263 (pNABC944) and $mip1^{exo-}$-CΔ263-mCherry (pNABC945) were constructed similarly to pNABC777 and pNABC943. The yeast genomic DNA used for amplifying the Mip1 sequence was from NABC573 ($mip1^{exo-}$).

### Mip1-G651S (pNABC946)
The plasmid containing mip1-G651S under pMip1 was generated by PCR amplification of the pMip1-mip1-G651S fragment (ending at Ser991 in Mip1) from wild-type yeast genomic DNA using primers AS271 and AS272. A stop codon was introduced at residue 991. The fragment was then Gibson cloned into pNABC555 between SacI and BamHI sites, removing the cyc1 promoter.

### Mip1-G651S-mNeongreen (pNABC947)
The plasmid containing mip1-G651S tagged with mNeongreen (pNABC947) was generated similarly. The pMip1-mip1-G651S fragment (ending at Ser991) was amplified using primers AS271 and AS272 from wild-type yeast genomic DNA. This fragment was Gibson cloned with a 17 amino acid linker followed by

mNeongreen (amplified using AS225 and AS192) and inserted into pNABC555 between SacI and BamHI sites.

### pNABC948 (Mip1-H734Y)

This plasmid, expressing mip1-H734Y under pMip1, was generated by PCR amplification of the pMip1-mip1-H734Y fragment (ending at Ser991 in Mip1) from wild-type yeast genomic DNA using primers AS273 and AS274. A stop codon was introduced at residue 991. The fragment was then Gibson cloned into pNABC555 between SacI and BamHI sites, removing the cyc1 promoter.

### pNABC949 (Mip1-H734Y-mNeongreen)

To generate the mNeongreen-tagged version of Mip1-H734Y (pNABC949), the pMip1-mip1-H734Y fragment (ending at Ser991 in Mip1) was amplified using primers AS273 and AS274 from wild-type yeast genomic DNA. This fragment was then Gibson cloned with a 10 amino acid linker followed by mNeongreen into pNABC555 between SacI and BamHI sites.All plasmids described were sequenced to confirm the presence of relevant sequences and/or mutations.

Plasmids for expression of the empty vector (pNABC557), mtDarT (pNABC561), Mip1-CΔ263 (pNABC775), $mip1^{exo-}$-CΔ263 (pNABC944), $mip1^{G651S}$ (pNABC946) and $mip1^{H734Y}$ (pNABC948) were freshly transformed into yeast cells before each experiment. This was to prevent mtDNA instability that might arise from the continuous maintenance and expression of both the toxin and Mip1 truncations/mutants.

## Yeast strain construction

Protrophic Cen.PK strain was used for all constructions.

### Mip1-mCherry (NABC772)

pNABC549 was used to amplify the mCherry-HygR cassette together with overhangs for Mip1 using primers AS79 and AS80. Transformants were screened and confirmed for Mip1 C-ter tagging using AS78 and AS81.

### Mip1-mNeongreen (NABC773)

pNABC553 was used to amplify the mNeonGreen-KanR cassette together with overhangs for Mip1 using primers AS79 and AS80. Transformants were screened and confirmed for Mip1 C-ter tagging using AS78 and AS81.

### $mip1^{exo-}$-mNeongreen (NABC680)

Same as NABC773, but in NABC573 cells.

### Δsml1 Mip1-mNeongreen (NABC766)

Same as NABC773, but in NABC639 cells.

### Cox4-mCherry (NABC569)

pNABC549 was used to amplify the mCherry-HygR cassette together with overhangs for Cox4 using primers ND328 and ND329. Transformants were screened and confirmed for Cox4 C-ter tagging using ND327 and AS81.

### Δsml1 (NABC639)

pNABC552 was used to amplify the HygR cassette together with overhangs for Sml1 using primers ND489 and ND490. Transformants were screened and confirmed for Δsml1 using ND488 and AS81.

### Δnuc1 (NABC641)

pNABC551 was used to amplify the KanR cassette together with overhangs for Nuc1 using primers AS92 and AS93. Transformants were screened and confirmed for Δnuc1 using AS91 and AS81.

### Δatg32 (NABC676)

pNABC552 was used to amplify the HygR cassette together with overhangs for Atg32 using primers ND162 and ND163. Transformants were screened and confirmed for Δatg32 using ND161 and AS81.

### Mip1 exonuclease-deficient mutant (NABC573)

Exonuclease-deficient Mip1 mutant ($mip1^{exo-}$) was constructed as previously described (Stuart et al, 2006; Medeiros et al, 2018). The mutant harbors two tandem mutations, D171A and E173A, in the ExoI motif of the exonuclease domain. The mutations were introduced by site-directed mutagenesis where the endogenous copy of Mip1 was deleted and subsequently transformed with a gene fragment containing Mip1 D171A, E173A double-mutation. Insertion of the mutant fragment was confirmed by sequencing. Deletion of Mip1 renders a cell ρ0. Therefore, mtDNA was reintroduced into $mip1^{exo-}$ mutants by crossing with wild-type cells of opposite mating-type. Haploid spores were dissected to obtain $mip1^{exo-}$ mutants with mtDNA.

## mtDNA damage induction

The protocol for inducing mtDNA damage was adopted from (Dua et al, 2022). Briefly, cells carrying empty vector control plasmid (pNABC557) or mtDarT plasmid (pNABC561), with NAT resistance gene, were transformed into yeast cell cells using the standard Lithium acetate protocol. Transformants with the plasmid were selected by plating on media containing Nourseothricin (NTC). Colonies were inoculated in YPG cultures containing NTC for plasmid maintenance. Upon reaching the early log-phase (OD ~0.2–0.4), samples were collected (0 h). Subsequently, the cultures were treated with 100 nM β-estradiol to induce expression. Inducer was maintained at all times during the experiment. Samples were collected at 4, 8 h post induction as indicated for specific experiments. Cells were maintained in mid-exponential phase growth throughout the experiment via back-dilution at appropriate time intervals.

## Survival assay

Survival assay to compare growth between different strains was performed by serially diluting log-phase cultures (starting from $OD_{600}$ of 0.06, diluted 10×, until $6 \times 10^{-9}$). In all, 6 μl of the diluted culture was spotted on agar plates with respective selection markers. The plates were incubated at 30 °C, and growth was monitored.

## Live cell staining of mtDNA

mtDNA was stained in live cells using SYBR Green I. The protocol was adopted from (Jajoo et al, 2016). In brief, 1 ml aliquots of cultures were taken at specified time points, pelleted and washed in 1× PBS. This was followed by resuspension of cells in 1× PBS followed by the addition of 0.5 μl of 10,000× SYBR Green I. After gently mixing the contents, cells were immediately spun down and washed with 1× PBS. The pellet was resuspended in an

appropriated volume of 1× PBS. In total, 1 µl of diluted culture was spotted on 1% water agarose pads for microscopy.

For DAPI staining, 1-ml cultures were spun down and the pellets were resuspended in 1× PBS. 0.5 µl of 10 mg/ml DAPI was added to the cell suspension above. The dye was resuspended gently into the cell pellet and the mixture was immediately spun down. The pellet was washed with 1× PBS, twice, before resuspending it in appropriate volume of 1× PBS. In all, 1 µl of diluted culture was spotted on 1% water agarose pads for microscopy.

## Western blot

Trichloroacetic acid (TCA) extraction method, as previously described, was used (Dua et al, 2022). Dilution and details of the antibodies can be found in Reagents and Tools table. The protein bands were detected using a chemiluminescence substrate-based assay. Separately, in each case a second protein gel was probed as loading control, and visualized with Coomassie brilliant blue staining solution. For quantification of relative proteins levels, band intensities of protein of interest were obtained from ImageJ. Values were normalized to the intensity of the loading control.

## G1/S arrest using α-factor

To arrest yeast cells in G1/S phase using α-factor (mating pheromone), Δbar1 (MATa) strain was used. Overnight cultures of cells expressing empty vector or mtDarT were grown to appropriate OD (early log phase, 0.2–0.4), after which they were pelleted. The cell pellet was carefully resuspended in a pre-warmed YPG medium containing α-factor (final concentration of 25 ng/ml, from 10 µg/ml stock solution in Methanol) and NTC. The culture was placed in an incubator for 3 h for arrest to occur completely. After 3 h, the majority of cells were arrested in G1/S phase as revealed by the formation of schmoo cells with oblong morphology (Rosebrock, 2017). Upon reaching this stage, inducer (β-estradiol) was added and samples were collected to monitor the mtDNA status. To maintain continuous arrest, cells were pelleted and resuspended in media containing fresh α-factor every 3 h.

## G2/M arrest using Nocodazole

Overnight cultures of cells expressing empty vector or mtDarT were grown to appropriate OD (early log phase, 0.2–0.4), after which they were pelleted. The cell pellet was carefully resuspended in pre-warmed YPG medium containing Nocodazole (final concentration of 9 µl/ml, from 1.5 mg/ml stock solution) and NTC. The culture was placed in an incubator for 3 h for arrest to occur completely. After 3 h, majority of cells were arrested in G2/M phase as revealed by the dumbbell shape of these cells (Rosebrock, 2017). Upon reaching this stage, inducer (β-estradiol) was added and samples were collected to monitor the mtDNA status. To maintain continuous arrest, cells were pelleted and resuspended in media containing fresh Nocodazole every 3 h.

## Flow cytometry

Cell cycle analysis using flow cytometry was performed following the protocol from (Haase and Reed, 2002). Cultures were collected and centrifuged at 3000 rpm, for 3 min at room temperature. After discarding the media, the cells were fixed in 70% ethanol overnight at 40 °C. Following fixation, cells were pelleted and then washed in 1 ml of 50mM Sodium citrate (pH 7.5). Cells were then resuspended in 500 µl RNAse A buffer (500 µl sodium citrate buffer + 5 µl RNase A) and incubated with shaking (350 rpm) at 50 °C for 2 h. Cells were then pelleted and resuspended in Proteinase K solution (20 µl enzyme (2 mg/ml) + 180 µl sodium citrate buffer) and incubated at 37 °C for 1 h, with shaking (350 rpm). Following Proteinase K treatment, cells were pelleted, washed and resuspended in 1ml of sodium citrate buffer. In total, 100 µl of this sample was made up to 1ml with sodium citrate buffer containing 1 µM SYTOX Green. The sample was sonicated on ice, at 20% amplitude for 10 s with two 5 s pulses with a gap of 5 s. Flow cytometry analysis was performed on Accuri C6+ flow cytometer.

## Multiple sequence alignment (MSA)

MSA of PolG and related orthologs was performed using COBALT: multiple alignment tool (Papadopoulos and Agarwala, 2007). The protein sequences were obtained from Uniprot (Bateman et al, 2023). Sequences are color-coded based on the ClustalX criteria (conservation and position) (Thompson et al, 1997). The displayed consensus sequence was obtained by setting the threshold at 50%.

## Fluorescence microscopy and image analysis

Fluorescence imaging of budding yeast cells was carried out on a wide-field, epifluorescence microscope (Nikon Eclipse Ti2-E), with motorized XY-stage, 60× plan apochromat objective (NA 1.41) and a pE-4000 light source (CoolLED). Focus was maintained using an infrared-based Perfect focus system. Z scanning was done using motorized Z-control. Z-stacks were acquired across 7µm of the cell in steps of 0.3µm. Images were acquired with Hamamatsu Orca Flash4.0 camera using NIS-elements software (version 5.01).

## Image analysis

Following the acquisition, image stacks were deconvolved in NIS-elements software using 3D-Lucy-Richardson method. A maximum projection of the deconvolved stacks was generated in ImageJ for fluorescence image analysis. The total number of nucleoids was obtained using the 3D-maxima finder plugin in ImageJ. Cell segmentation was performed using Cell-ACDC program (Padovani et al, 2022) generating cell ROI's for single cell-based analyses. ROI's with incorrect segmentation or with abnormal SGI staining were manually excluded from the analysis. A number of nucleoids per cell was obtained by custom macros written using the Mitochondria analyser plugin (Chaudhry et al, 2020) in ImageJ where the number of foci was overlayed on cell ROI's to obtain number of nucleoid foci per cell. Cell area was calculated using Analyse Particles function in ImageJ. Correlation density plots were plotted using RStudio (Version 1.3.1093).

For colocalization analysis, line profiles were created on composite images using ImageJ to generate intensity values. The values obtained for each channel was overlayed and plotted using GraphPad Prism (8.4.2) software. Quantification of the percentage of cells with nucleoids was done using mother cells alone to avoid inconsistencies that might arise from incomplete mtDNA segregation in bud cells. Correlation plots were plotted ignoring the zero

values for ρ⁰ cells and density plots for mt-nucleoid number scaling per cell were plotted for all the cells (inc. ρ⁰) in a field. For these plots, all the cells (mother, bud) in the image field were used.

All microscopy data are deposited in the Biostudies Portal (accession number S-BIAD1418).

## Statistical analysis

Statistical analyses were performed using GraphPad Prism (8.4.2) after assessing normality using the Shapiro–Wilk test. For comparisons between two groups, parametric two-tailed unpaired $t$ tests were applied for normally distributed data, while nonparametric Wilcoxon's tests were applied for non-normal data. To compare differences in percentages of cells with nucleoid foci across multiple time points, a repeated measures one-way ANOVA was employed. This was followed by use of Tukey's multiple comparisons test to derive exact $P$ values for differences between two groups of data. Since the percentages for empty vector control were nearly 100%, nonparametric tests were applied to account for lack of variance. For assessing correlations, Spearman's correlation coefficient was obtained through custom scripts on R. $P$ values $\leq 0.05$ were considered to be significant, $P$ values $\leq 0.05$ were indicated by (*), $P$ values $\leq 0.01$ by (**), P values $\leq 0.001$ by (***) and $P$ values $\leq 0.0001$ by "****". $P$ values $> 0.05$ were considered non-significant. Details of the test for each dataset including results obtained can be accessed in Dataset EV1.

## Data availability

All the source data from this publication have been deposited in the Biostudies Portal and can be accessed via https://www.ebi.ac.uk/biostudies/bioimages/studies/S-BIAD1418.

The source data of this paper are collected in the following database record: biostudies:S-SCDT-10_1038-S44319-025-00380-1.

## Peer review information

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

## Acknowledgements

The authors are grateful to Dr. Nitish Dua and Prof. Sunil Laxman for sharing of reagents. The authors thank Dr. Nitish Dua, Dr. Asha Mary Joseph and Mr. Aditya Kamat for helpful discussions and feedback on the manuscript. The authors would like to thank Dr. Francesco Padovani for assistance with Cell-ACDC customizations. The authors would like to thank Mr. Anton Iyer for help with coding. This work was supported by fellowships from CSIR (AS) as well as grants from HFSP RPG (AB) (RGP0038/2021-102) and intramural funding from NCBS-TIFR (AB) (Department of Atomic Energy, Government of India, Project Identification No. RTI 4006).

## Author contributions

**Akshaya Seshadri**: Conceptualization; Resources; Data curation; Formal analysis; Validation; Investigation; Visualization; Methodology; Writing—original draft; Writing—review and editing. **Anjana Badrinarayanan**: Conceptualization; Data curation; Supervision; Funding acquisition; Validation; Investigation; Writing—original draft; Project administration; Writing—review and editing.

Source data underlying figure panels in this paper may have individual authorship assigned. Where available, figure panel/source data authorship is listed in the following database record: biostudies:S-SCDT-10_1038-S44319-025-00380-1.

## Disclosure and competing interests statement

The authors declare no competing interests.

# Expanded View Figures

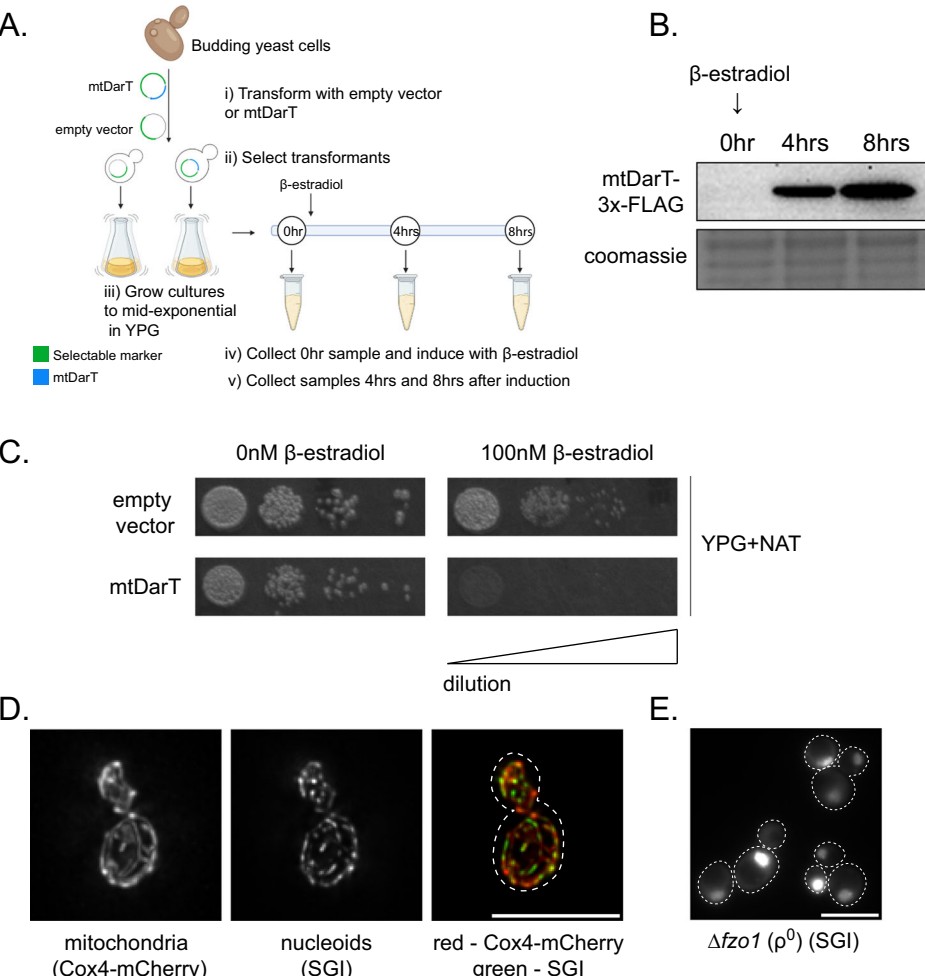

**Figure EV1. mtDNA damage results in mtDNA clearance.**

(A) Experimental setup followed for inducing mtDNA damage using the mtDarT system is shown (Dua et al, 2022). Cells were grown in a non-fermentable carbon source (YPG) and cultures were maintained in mid-exponential growth during the course of the entire experiment. OD600 at the time of imaging was always ensured to be comparable between control (empty vector) and mtDarT-treated cells. Schematic created with Biorendor.com. (B) Representative Western blot of mtDarT-3x-FLAG cells before (0 h) and after (4, 8 h) 100nM β-estradiol addition. Whole cell lysate is probed with Coomassie as loading control ($n = 3$ independent repeats). (C) Survival of yeast cells with empty vector or mtDarT plasmid, grown on YPG media with and without 100 nM β-estradiol. Representative image from three independent repeats is shown. Scale bar refers to the increasing dilution of cells from the left spot to the right-most spot. (D) Representative images of nucleoids stained with SGI and mitochondria marked with Cox4-mCherry. Dashed lines represent cell boundaries. (E) SGI staining in Δfzo1 cells (ρ⁰, mtDNA absent). Dashed lines represent cell boundaries. Scale bar, 8 μm here, and in all other images. Source data are available online for this figure.

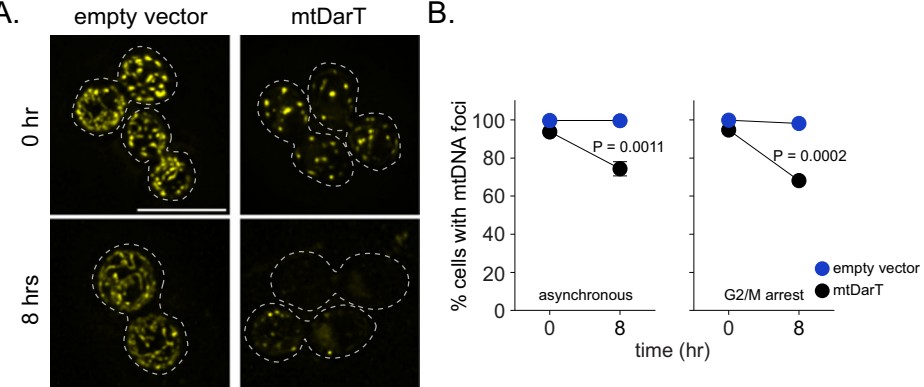

**Figure EV2.   mtDNA loss occurs independent of cell cycle progression.**

(A) Representative images of nucleoid foci in G2/M-arrested, empty vector and mtDarT-expressing cells at 0 h (top) and 8 h after induction (bottom). Dashed lines represent cell boundaries. (B) Percentage of cells containing mtDNA foci in asynchronous (left) or G2/M-arrested (right) empty vector and mtDarT cells at 0 and 8 h after induction. Data shown from three independent repeats (*n* >138 cells per group, per repeat). Mean and SD are shown. Significance was calculated using Unpaired *t* test (two-tailed). **$P \leq 0.01$, ***$P \leq 0.001$. Scale bar, 8 μm here, and in all other images. Source data are available online for this figure.

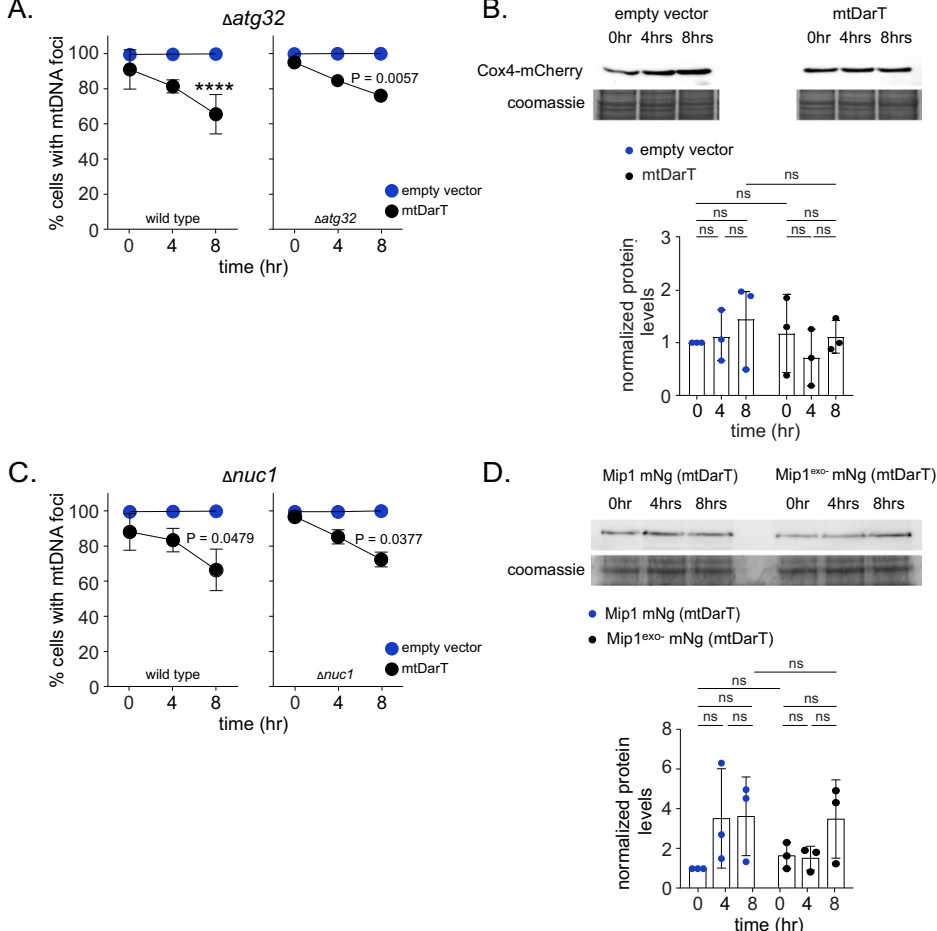

Figure EV3.   mtDNA loss is dependent on Mip1 exonuclease.

(**A**) Percentage of cells with mtDNA foci in wild-type (left) or Δ*atg32* cells (right) at 0, 4 and 8 h after induction with empty vector (blue) and mtDarT (black). Data shown from three independent repeats (*n* >144 cells per group, per repeat). Mean and SD are shown. Significance was calculated using repeated measures one-way ANOVA and post hoc tests. **$P \leq 0.01$, ****$P \leq 0.0001$. (**B**) Western blot of Cox4-mCherry in empty vector and mtDarT cells, before (0 h) and after (4, 8 h) damage induction. Representative western blot image is shown at the (top) and levels are quantified at the (bottom). $n = 3$ independent repeats. Mean and SD are shown. Significance was calculated using repeated measures one-way ANOVA and post hoc tests. (**C**) Percentage of cells with mtDNA foci in wild-type (left) or Δ*nuc1* cells (right) at 0, 4 and 8 h after induction with empty vector (blue) and mtDarT (black). Data shown from three independent repeats (*n* >115 cells per group, per repeat). Mean and SD are shown. Significance was calculated using repeated measures one-way ANOVA and post hoc tests. *$P \leq 0.05$. (**D**) Western blot of Mip1-mNeonGreen and Mip1exo--mNeonGreen before (0 h) and after (4, 8 h) damage induction. Representative western blot image is shown at the (top) and levels are quantified at the (bottom). $n = 3$ independent repeats. Mean and SD are shown. Significance was calculated using repeated measures one-way ANOVA and post hoc tests. Source data are available online for this figure.

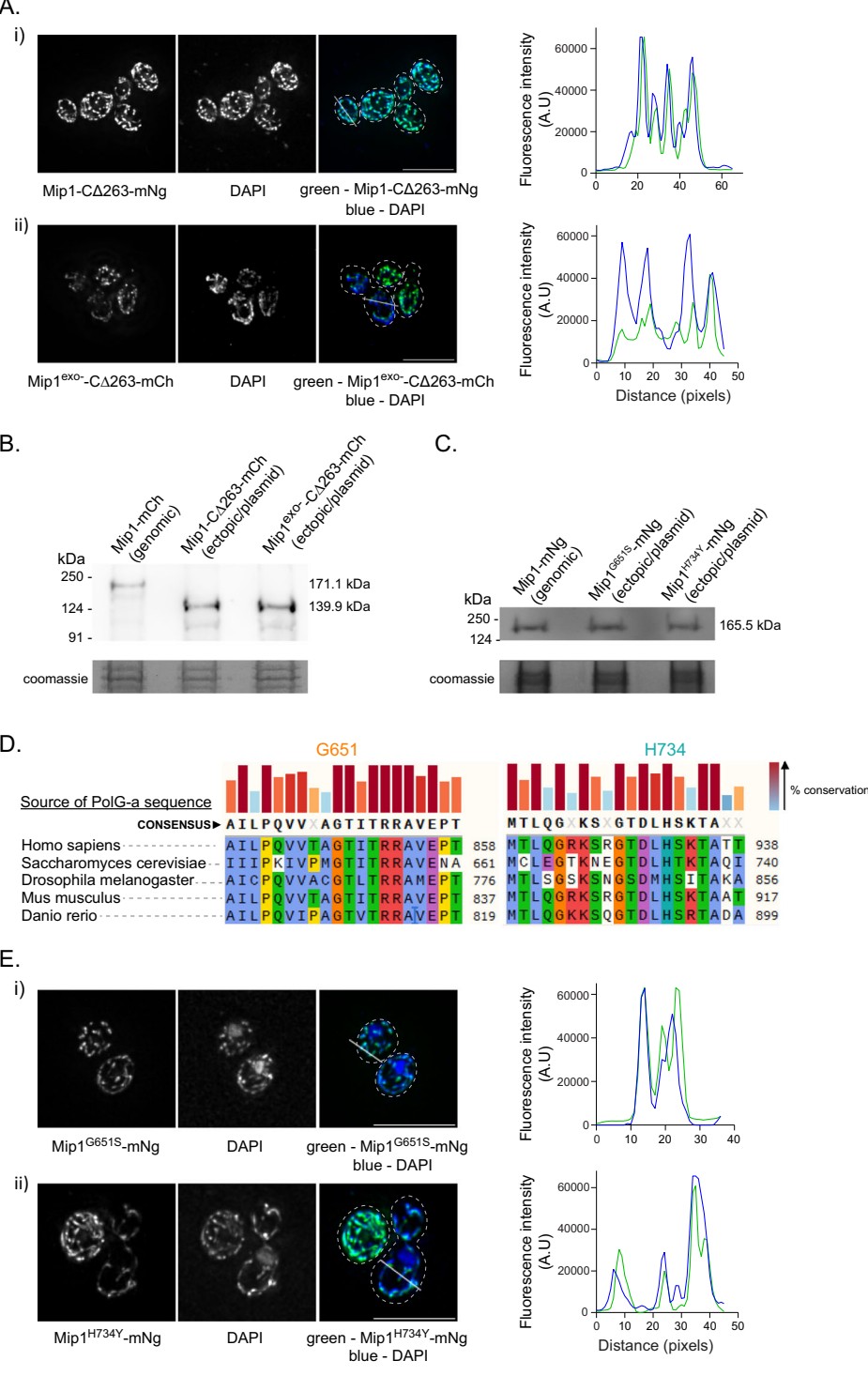

◀

**Figure EV4. Exonuclease activity of Mip1 drives mtDNA loss under damage.**

(A) Left: localization of Mip1-CΔ263-mNeonGreen (i) or *mip1^exo-*-CΔ263-mCherry (ii) and mt-nucleoids stained with DAPI in *mip1^exo-* cells. Right: line profiles of Mip1-CΔ263-mNeonGreen (i) or Mip1^exo-*-CΔ263-mCherry (ii) and mt-nucleoids stained with DAPI in *mip1^exo-* cells. Line profile for a representative across an ROI is shown (white line in merged panel). Dashed lines represent cell boundaries. (B) Western blot of Mip1-mCherry, Mip1-CΔ263-mCherry and Mip1^exo-*-CΔ263-mCherry. Representative image with loading control (Coomassie) is shown from one of the three independent repeats. (C) Western blot of Mip1-mNeonGreen, Mip1^G651S*-mNeonGreen and Mip1^H734Y*-mNeonGreen. Representative image with loading control (Coomassie) is shown from one of the three independent repeats. (D) Multiple sequence alignment of Human PolG and related orthologues, including Mip1, performed using COBALT to show the conservation and position of G651 and H734 across organisms. (E) Left: localization of Mip1^G651S*-mNeonGreen (i) or Mip1^H734Y*-mNeonGreen (ii) and mt-nucleoids stained with DAPI in *mip1^exo-* cells. Right: line profiles of Mip1^G651S*-mNeonGreen (i) or Mip1^H734Y*-mNeonGreen (ii) and mt-nucleoids stained with DAPI in *mip1^exo-* cells. Line profile for a representative across an ROI is shown (white line in merged panel). Dashed lines represent cell boundaries. Scale bar, 8 µm here, and in all other images. Source data are available online for this figure.

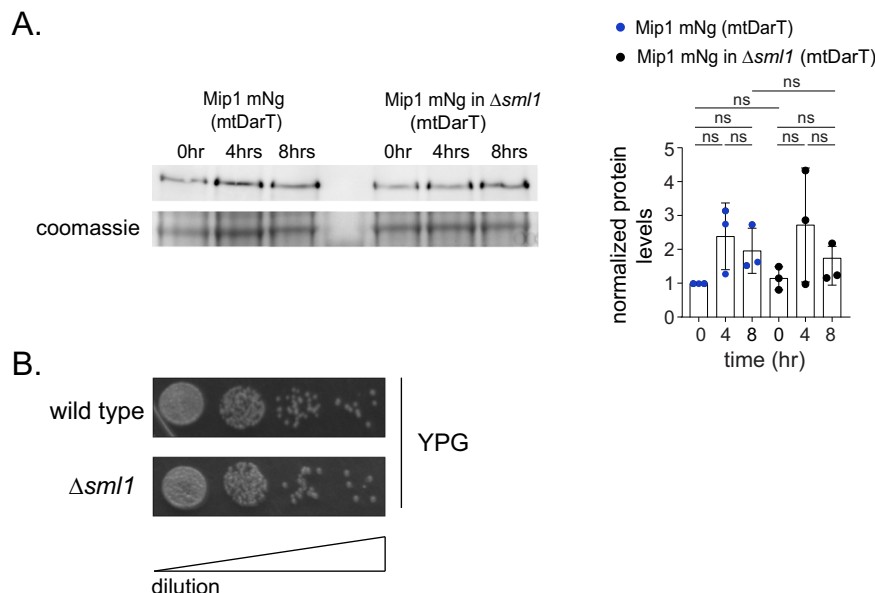

**Figure EV5. Rates of mtDNA loss are influenced by dNTP levels.**

(A) Western blot of Mip1-mNeonGreen in wild-type and Δ*sml1* cells before (0 h) and after (4, 8 h) damage induction. Representative western blot image is shown on the (left) and levels are quantified on the (right). *n* = 3 independent repeats. Mean and SD are shown. Significance was calculated using repeated measures one-way ANOVA and post hoc tests. (B) Serial dilution growth assay to measure growth of Δ*sml1* cells in comparison to a wild-type control. Representative image from three independent repeats is shown. Scale bar refers to the increasing dilution of cells from the left spot to the right-most spot. Source data are available online for this figure.

