## [Peer Review File · EMBO Reports]

Exonuclease action of replicative polymerase gamma drives damage-induced mitochondrial DNA clearance

Akshaya Seshadri and Anjana Badrinarayanan

Corresponding author(s): Anjana Badrinarayanan (anjana@ncbs.res.in)

Review Timeline:

Transfer Date:	7th May 24
Editorial Decision:	15th May 24
Revision Received:	21st Oct 24
Editorial Decision:	13th Dec 24
Revision Received:	17th Dec 24
Accepted:	20th Dec 24

Editor: Deniz Senyilmaz Tiebe

Transaction Report: This manuscript was transferred to EMBO reports following peer review at The EMBO Journal.

Referee #1:

The Badrinarayanan lab has previously reported that heterologous expression of a bacterial toxin, DarT, targeted to mitochondria induces damage in mitochondrial DNA in yeast. In the present study the authors report that mtDNA nucleoids are actively degraded upon expression of mtDarT. They convincingly show that mtDNA degradation depends on the exonuclease activity of the mitochondrial DNA polymerase, Mip1, and is independent of the cell cycle. Furthermore, they provide evidence that efficient mtDNA degradation requires both the polymerase and exonuclease domains of Mip1 and is influenced by cellular dNTP levels. In sum, the authors report some important and interesting findings. However, mechanistic insights of the study are rather limited and the manuscript falls short of demonstrating that mtDNA clearance occurs in a similar way in more physiological DNA damage scenarios.

Specific points

1. The statement that maintenance of mtDNA is essential for cell survival (line 24) is not correct for yeast, at least on fermentable carbon sources. The information that cells were grown on non-fermentable carbon sources is hidden in the methods section. Please clearly state this in the main text and/or figure legends.
2. Why does S1D staining result in nuclear staining in rho0 cells, but not in wild type? Has this been described before, or is there any explanation?
3. The y-axis of graphs indicating loss of mtDNA nucleoids is labeled "percentage of rho+ cells". This would imply that mtDNA is intact and cells are respiratory competent. Has this been checked?
4. The authors induced mtDarT expression for up to 8 hours. This corresponds to several generation times. They should check how mtDarT expression affects growth under these conditions, and they should avoid comparing stationary cells mtDarT with logarithmically growing WT cells.
5. According to the Western blot shown in Fig. S4 truncated Mip1 constructs are highly overexpressed. Can the authors exclude that the expression levels affect the results shown in Figure 4?
6. Western blots showing expression levels of Mip1 variants or of Mip1 in different strain backgrounds should always show wild type controls in direct comparison. Also, the experiments performed in delta sml1 (Fig. 5) have to be shown in direct comparison with wild type SML1.
7. The manuscript contains some spelling and grammatical errors and would profit from proofreading.

Referee #2:

This paper explores the role of the Mip1 exonuclease in the clearance of damaged mtDNA. Using a mitochondrial targeted DarT that ribosylates the mitochondrial single stranded DNA, the authors demonstrate that this causes a loss of mtDNA that is cell cycle independent. Exploring the role of the Mip1 exonuclease revealed that this loss is dependent on the exonuclease function and influenced by dNTP levels. The authors conclude that upon encountering DNA damage, the Mip1 polymerase switches from DNA replication mode to DNA derogation using the exonuclease function. The conclusions are intriguing and should represent a substantial understanding to the

clearance of damaged mtDNA genomes. However, as presented, there are several additional experiments or considerations to make before this can be considered.

Comments:

Figure 1 - this experiment demonstrates the reduction of mtDNA after induction of the mtDarT. Based upon the results the authors conclude that induction of DNA damage causes loss of mtDNA through a clearance mechanism. However, this does not exclude the possibility that the damage is simply inhibiting mtDNA replication causing a decrease in the copy number, as opposed to mtDNA degradation. The experiment in Fig 3 partially address this, but these experiments should be done side by side.

The authors propose that the mtDarT generated adducts triggers exonucleolytic digestion of the mtDNA as evidence by the loss of mtDNA. However, there is no direct experiment presented that demonstrates degradation of the mtDNA through this process. The decline in mtDNA could be done in a mitophagy mediated degradation of other means of clearance of the damaged mtDNA not involving degradation. Experiments are needed to see degradation products such as following the fate of the mtDNA after induction by the mtDarT (in Wt and Mip1exo-) by such methods as PCR, Southern blot, and sequencing.

It is unclear whether the two deletion mutants even produce a stably folded protein. To demonstrate that these two deletion of the Mip1 protein are even functional, the authors need to purify the recombinant protein and demonstrate that they still retain exonuclease activity in vitro. I suspect that these proteins are not stable in vitro and are being degraded. The Western blot in Fig S4 does not demonstrate that the protein is stable and retains exonuclease activity.

The model proposed that when Mip1 encounters a DNA lesion, it switches from replication to degradation can be easily tested in vitro with purified proteins and labeled adducted primer template.

In addition to the use of the two deletion mutants, the authors should consider a point mutation in the polymerase active site that decreases nucleotide insertions without compromising cell survival on glycerol. The literature has several examples of homologous human disease mutations that have been studied in yeast that cause more sensitivity of single stranded DNA to damaging agents.

Referee #3:

The manuscript poses intriguing ideas and it is truly a shame that the manuscript does not fulfill scientific standards: unless I missed critical explanations, fundamental flaws in the experimental setup prevent any support for publication from my side:

(1) WT controls are not present or not run in parallel with mutant strains in figures 3, 4, and 5B-C as well as S3A-B and S5. Thus, it is impossible to compare the data and the effects of the mutations. Empty vector controls by themselves are insufficient to assess the effects of the mutations in comparison to WT cells. Unless the experiments in the mentioned figure are repeated with the inclusion of the required controls run in parallel, these data are not publishable. Given these crucial problems, I will not provide a detailed review for this manuscript.

(2) The authors published previously that the expression of a mip1-exo- variant would accelerate mtDNA loss upon mtDarT expression compared with wildtype MIP1 expressing cells (Dua et al., JCB 2022). However, these data are not even mentioned or discussed in the current manuscript. Why do the authors observe opposite effects in these experiments?

Editor comments –

The discrepancy with your earlier work needs to be clarified (referee #3, point 2).

> We apologise that our writing was not clear in this regard, and thank the editor and reviewer for bringing this point up. Our earlier work on characterizing mtDarT (Dua et al., 2022) was done in growth conditions with a fermentable carbon source. Hence yeast cells can continue to grow and divide even in the absence of functional mitochondria/ mtDNA. In this study too, we had shown that Mip1 exonuclease activity played a role in mtDNA clearance; in its absence, clearance rates were exacerbated via asymmetric mtDNA segregation and cell division, resulting in rapid generation of rho0 cells. Indeed, when cell cycle was arrested in *mip1^{exo-}* cells growing in YPD, we did not observe mtDNA loss (PMID: 36074064, Fig. 5F). In our current work, we specifically studied the role of Mip1 in mtDNA clearance in a non-fermentable carbon source, so as to remove the potential effects of cell division in contributing to mtDNA clearance (as in this growth condition cells would need functional mitochondria/ mtDNA to survive). Hence, there is no discrepancy between our earlier and present work. In fact, both studies lend support to the idea that cells require Mip1 for mtDNA clearance under damage. In case of growth on fermentable carbon source, yeast cells utilize cell division as an additional means to drive this clearance (specifically when Mip1 exonuclease is compromised, Dua et al., 2022). When the possibility of viable cell divisions is compromised (non-fermentable carbon source), Mip1 exonuclease appears to be the sole driver of this clearance during mtDNA stress.

We realize that we should have highlighted this more clearly in our manuscript. We propose to include a section in the discussion synthesizing evidences from both studies (Dua 2022 and present study) to highlight the role of Mip1 and the importance of mtDNA loss under conditions of mtDNA damage.

- The switching from replication to degradation model requires additional support by employing in vitro approaches (referee #2, comment starting as 'The model proposed...') and repeating the experiments by including the controls requested by referee #3 (point 1).

> The focus of the present work was to track the fate of mtDNA in vivo under a perturbation resembling a replication-stalling lesion. Given this, we have largely adopted genetic strategies to perturb various pathways and Mip1, and correspondingly quantify changes to the mtDNA numbers in vivo. Our hypothesis regarding Mip1 exonuclease activity in the model comes from detailed studies on Mip1/PolG and other polymerases in vitro wherein concepts such as stimulation of exonuclease activity in response to replication hurdles, synergistic interactions between different Mip1/PolG mutations in trans, coupling between exonuclease and polymerase activities of the enzyme etc. have been shown biochemically. These studies include (PMID: 32037587; PMID: 22432028, PMID: 20601675 and PMID: 31445096).

Our proposed model incorporates the observations from these studies and our own to explain loss of entire genome sized copies in vivo. The consequence of which is a mechanism to rapidly clear damaged DNA copies by the proofreading function of the polymerase. In addition, we tested for potential factors stimulating the Mip1 exonuclease within the mitochondria (such as dNTP levels). Indeed, new in vitro studies (in light of our observations) would be needed to dissect the detailed molecular mechanism of Mip1 exonuclease activity. Such a detailed study requires significant time and effort, and this is outside the scope and focus of our present work. We do plan to pursue this avenue in the future as a dedicated project.

To ensure that our in vivo observations remain the central focus of our present work, we propose to tone-down any discussions related to the mechanistic aspects of Mip1 activity. We agree with the request for wild type controls from reviewer 3. We will conduct these experiments and re-plot re graphs with individual wild type data for comparison.

- Contribution of mtDNA replication inhibition to the decrease in mtDNA levels needs to be ruled out (referee #2, comment starting as Figure 1...).

> While replication may also be compromised under conditions of DNA damage, our data strongly support a major role for exonuclease action in the observed mtDNA loss. In case of Mip1, pol and exo domains are in the same protein and studies show that if Pol activity is compromised, this can increase the exo activity (PMID: 22114710, PMID: 22432028). In our present study, using the *mip1^{exo-}* mutant we show that pol is still active, albeit compromised likely due to increased basal error rate. Thus, in an *mip1^{exo-}* mutant where polymerase activity is still present, cells do not lose mtDNA under damage. This highlights that a possible perturbation of replication under damage does not immediately reflect as mtDNA loss, at least in the timescales in which we track mtDNA. This has also been seen in other contexts (starvation (PMID: 29519802) and linearized mtDNA loss (PMID: 29712893)).

We propose to repeat analysis in figure 3 with the wild type data (as suggested by the reviewer). In addition, we propose to carry out qPCR experiments to assess mtDNA loss via another assay. In the discussion section, we will provide a balanced view of potential mechanisms for mtDNA loss, and will include possible roles for compromised mtDNA replication and mitophagy (in metazoans; see next point). Based on editor advise, we can also carry out EdU labelling experiments to measure mtDNA replication in the presence and absence of mtDNA damage.

- The fate of degraded mtDNA needs to be better investigated to exclude the contribution of mitophagy (referee #2, comment starting as 'The authors propose that...')

> We did investigate a role for mitophagy in our experiments. We observed significant mtDNA loss under damage in cells deleted for Atg32. Furthermore, we did not observe a significant change in mitochondrial volume, or Cox4 levels under damage conditions. These observations are also consistent with our previous findings that mitophagy (which is typically observed in much longer timescales) may not play a role under mtDNA damage in yeast (Dua et al., 2022, PMID: 36074064).

We propose to include the above experiments in our revised manuscript. In addition, we propose to carry out qPCR measurements of mtDNA loss under damage conditions.

- Further evidence needs to be presented to demonstrate functionality of the deletion mutants (referee #2, comment starting as 'It is unclear whether...'). Alternatively, the analyses may be repeated by employing point mutants located in the active sites (referee #2, comment starting as 'In addition to the use of...').

> We thank the reviewer and editor for highlighting this issue and for the excellent suggestions. The $\Delta 263$ mutant is a well-characterized mutant and has already been shown in vitro to have exonuclease activity. We acknowledge that the $\Delta 669$ mutant is not characterized in this manner.

We are happy to remove data and reference to the $\Delta 669$ mutant, as it does not add substantially to the main conclusions of this section. Instead, we propose to make active site mutants as suggested by the reviewer and editor to further support this section of the manuscript. We will also introduce the exonuclease mutations in the $\Delta 263$ background to further validate that the mtDNA loss observed in this mutant was due to exonuclease activity. The biochemical characterizations of the $\Delta 669$ mutant are outside the scope of our current study.

Dear Anjana,

Thank you for transferring your manuscript to EMBO Reports along with referee reports from another venue and sending your preliminary point-by-point response. I have now read your points carefully. I appreciate that you can address many of the concerns raised and see that the proposed experiments will strengthen the manuscript.

Having looked at everything, I would like to invite you to submit a revised manuscript as in your revision plan. However, I would like to point out that we need strong support from the referees to consider publication here.

Please revise your manuscript with the understanding that the referee concerns (as in their reports) must be fully addressed and their suggestions taken on board. Please address all referee concerns in a complete point-by-point response. Acceptance of the manuscript will depend on a positive outcome of a second round of review. It is EMBO reports policy to allow a single round of major experimental revision only and acceptance or rejection of the manuscript will therefore depend on the completeness of your responses included in the next, final version of the manuscript.

We realize that it is difficult to revise to a specific deadline. In the interest of protecting the conceptual advance provided by the work, we recommend a revision within 3 months. Please discuss the revision progress ahead of this time with me if you require more time to complete the revisions, or if you have questions or comments regarding the revision (also by video chat).

1. A data availability section providing access to data deposited in public databases is missing (where applicable).
2. Your manuscript contains statistics and error bars based on $n=2$. Please use scatter plots in these cases.

You can submit the revision either as a Scientific Report or as a Research Article. For Scientific Reports, the revised manuscript can contain up to 5 main figures and 5 Expanded View figures, and it should not exceed 27000 characters. If the revision leads to a manuscript with more than 5 main figures it will be published as a Research Article. In this case the Results and Discussion section should be separate. If a Scientific Report is submitted, these sections have to be combined. This will help to shorten the manuscript text by eliminating some redundancy that is inevitable when discussing the same experiments twice. In either case, all materials and methods should be included in the main manuscript file.

4) a .docx formatted letter INCLUDING the reviewers' reports and your detailed point-by-point responses to their comments. As part of the EMBO publication's Transparent Editorial Process, EMBO reports publishes online a Review Process File (RPF) to accompany accepted manuscripts. This File will be published in conjunction with your paper and will include the referee reports, your point-by-point response and all pertinent correspondence relating to the manuscript.

<https://www.embopress.org/page/journal/14693178/authorguide#transparentprocess>

You are able to opt out of this by letting the editorial office know (emboreports@embo.org). If you do opt out, the Review Process File link will point to the following statement: "No Review Process File is available with this article, as the authors have

chosen not to make the review process public in this case."

5) a complete author checklist, which you can download from our author guidelines

<https://www.embopress.org/page/journal/14693178/authorguide>. Please insert information in the checklist that is also reflected in the manuscript. The completed author checklist will also be part of the RPF.

6) Please note that all corresponding authors are required to supply an ORCID ID for their name upon submission of a revised manuscript (<<https://orcid.org/>>). Please find instructions on how to link your ORCID ID to your account in our manuscript tracking system in our Author guidelines

<<https://www.embopress.org/page/journal/14693178/authorguide#authorshipguidelines>>

7) Before submitting your revision, primary datasets produced in this study need to be deposited in an appropriate public database (see <https://www.embopress.org/page/journal/14693178/authorguide#datadeposition>). Please remember to provide a reviewer password if the datasets are not yet public. The accession numbers and database should be listed in a formal "Data Availability" section placed after Materials & Method (see also

<https://www.embopress.org/page/journal/14693178/authorguide#datadeposition>). Please note that the Data Availability Section is restricted to new primary data that are part of this study. * Note - All links should resolve to a page where the data can be accessed. *

Additional information on source data and instruction on how to label the files are available:

<https://www.embopress.org/page/journal/14693178/authorguide#sourcedata>

9) Our journal encourages inclusion of *data citations in the reference list* to directly cite datasets that were re-used and obtained from public databases. Data citations in the article text are distinct from normal bibliographical citations and should directly link to the database records from which the data can be accessed. In the main text, data citations are formatted as follows: "Data ref: Smith et al, 2001" or "Data ref: NCBI Sequence Read Archive PRJNA342805, 2017". In the Reference list, data citations must be labeled with "[DATASET]". A data reference must provide the database name, accession number/identifiers and a resolvable link to the landing page from which the data can be accessed at the end of the reference. Further instructions are available at <http://www.embopress.org/page/journal/14693178/authorguide#referencesformat>

10) Regarding data quantification (see Figure Legends:

<https://www.embopress.org/page/journal/14693178/authorguide#figureformat>)

12) Please also note our reference format:

13) All Materials and Methods need to be described in the main text. We would encourage you to use 'Structured Methods', our new Methods format. According to this format, the Methods section should include a Reagents and Tools Table (listing key reagents, experimental models, software and relevant equipment and including their sources and relevant identifiers) followed by a Methods and Protocols section in which we encourage the authors to describe their methods using a step-by-step protocol format with bullet points, to facilitate the adoption of the methodologies across labs. More information on how to adhere to this format as well as downloadable templates (.doc or .xls) for the Reagents and Tools Table can be found in our author guidelines: <<https://www.embopress.org/page/journal/14693178/authorguide#manuscriptpreparation>>.

An example of a Method paper with Structured Methods can be found here:
<<https://www.embopress.org/doi/10.15252/msb.20178071>>.

I look forward to seeing a revised version of your manuscript when it is ready. Please let me know if you have questions or comments regarding the revision.

Kind regards,

Deniz

Deniz Senyilmaz Tiebe, PhD
Scientific Editor
EMBO Reports

Referee #1:

The Badrinarayanan lab has previously reported that heterologous expression of a bacterial toxin, DarT, targeted to mitochondria induces damage in mitochondrial DNA in yeast. In the present study the authors report that mtDNA nucleoids are actively degraded upon expression of mtDarT. They convincingly show that mtDNA degradation depends on the exonuclease activity of the mitochondrial DNA polymerase, Mip1, and is independent of the cell cycle. Furthermore, they provide evidence that efficient mtDNA degradation requires both the polymerase and exonuclease domains of Mip1 and is influenced by cellular dNTP levels. In sum, the authors report some important and interesting findings. However, mechanistic insights of the study are rather limited and the manuscript falls short of demonstrating that mtDNA clearance occurs in a similar way in more physiological DNA damage scenarios.

Specific points

1. The statement that maintenance of mtDNA is essential for cell survival (line 24) is not correct for yeast, at least on fermentable carbon sources. The information that cells were grown on non-fermentable carbon sources is hidden in the methods section. Please clearly state this in the main text and/or figure legends.

> We apologise for the lack of clarity in our writing. We have now corrected L24 to reflect mtDNA essentiality under respiratory growth conditions for yeast. Additionally, we have clearly stated our growth conditions in the first section of the results (L105-L115) as well as the first figure legend where the experimental setup is introduced (L695-L696, L777-L780).

2. Why does S1D staining result in nuclear staining in rho0 cells, but not in wild type? Has this been described before, or is there any explanation?

> This has been reported previously as well (PMID: 31599702, PMID: 1324172, PMID: 19563757, PMID: 25170845). It is proposed that during pulse-staining with nucleoid stains such as DAPI/ Sybr, the stain would be taken up by mtDNA readily and preferentially, and hence it is predominantly visible when mtDNA is present in the cell. In the absence of mtDNA, the stain is now taken up by the nucleus instead and is hence visible.

3. The y-axis of graphs indicating loss of mtDNA nucleoids is labeled "percentage of rho+ cells". This would imply that mtDNA is intact and cells are respiratory competent. Has this been checked?

> We thank the reviewer for raising this point. To avoid confusion, we have re-plotted relevant graphs with the y-axis reflecting (and labelled) '% cells with mtDNA foci'.

4. The authors induced mtDarT expression for up to 8 hours. This corresponds to several generation times. They should check how mtDarT expression affects growth under these conditions, and they should avoid comparing stationary cells mtDarT with logarithmically growing WT cells.

> In our experimental setup, we ensure that cells do not enter stationary phase via appropriately back-diluting and maintaining all cultures in mid-log phase. Furthermore, imaging is carried out at comparable log-phase OD values for all samples (including empty vector controls).

We have stated the same in the results (L109-L115) and methods (L368-L369) sections of the revised manuscript.

5. According to the Western blot shown in Fig. S4 truncated Mip1 constructs are highly overexpressed. Can the authors exclude that the expression levels affect the results shown in Figure 4?

> So as to rule out any overexpression-associated effects, we introduced the exonuclease mutation in the truncation construct. Western blot of both the Mip1-C Δ 263 and exonuclease deficient Mip1-C Δ 263 show similar expression levels, albeit more than the endogenous Mip1 protein (revised Fig. EV4A-B). If expression levels alone contributed to the observed loss in mtDNA foci, then overexpression of an exonuclease-deficient Mip1-C Δ 263 should also result in mtDNA loss. However, we do not observe such mtDNA loss occurring in the absence of exonuclease activity (revised Fig. 4B-C). Based on these results, we conclude that the protein levels alone do not contribute to mtDNA loss when these constructs are ectopically expressed.

These results are now included in Fig. (4 and EV4) and presented in results section from L240-L246 of the revised manuscript.

6. Western blots showing expression levels of Mip1 variants or of Mip1 in different strain backgrounds should always show wild type controls in direct comparison. Also, the experiments performed in delta sml1 (Fig. 5) have to be shown in direct comparison with wild type SML1.

> We have re-plotted the graphs and provided western blots with wild type controls for direct comparison in revised figures (EV3C, EV4B, EV4C, EV5A). The conclusions drawn from these new data are the same as reported in our original submission.

7. The manuscript contains some spelling and grammatical errors and would profit from proofreading.

> We apologise for this issue, and have corrected the same.

Referee #2:

This paper explores the role of the Mip1 exonuclease in the clearance of damaged mtDNA. Using a mitochondrial targeted DarT that ribosylates the mitochondrial single stranded DNA, the authors demonstrate that this causes a loss of mtDNA that is cell cycle independent. Exploring the role of the Mip1 exonuclease revealed that this loss is dependent on the exonuclease function and influenced by dNTP levels. The authors conclude that upon encountering DNA damage, the Mip1 polymerase switches from DNA replication mode to DNA derogation using the exonuclease function. The conclusions are intriguing and should represent a substantial understanding to the clearance of damaged mtDNA genomes. However, as presented, there are several additional experiments or considerations to make before this can be considered.

Comments:

1. Figure 1 - this experiment demonstrates the reduction of mtDNA after induction of the mtDarT. Based upon the results the authors conclude that induction of DNA damage causes loss of mtDNA through a clearance mechanism. However, this does not exclude the possibility that the damage is simply inhibiting mtDNA replication causing a decrease in the copy number, as opposed to mtDNA degradation. The experiment in Fig 3 partially addresses this, but these experiments should be done side by side.

> While replication may also be compromised under conditions of DNA damage, our data strongly support a major role for exonuclease action in the observed mtDNA loss. In our present study, the *mip1^{exo-}* mutant still has an active polymerase. However, these cells do not lose mtDNA upon damage induction, suggesting that exonuclease activity is indeed the primary contributor to the observed phenotype (Fig. 3C). This highlights that a possible perturbation of replication under damage does not immediately reflect as mtDNA loss, at least in the timescales in which we track mtDNA. This has also been reported in other contexts (starvation (PMID: 29519802) and linearized mtDNA loss (PMID: 29712893)).

As per the reviewer's suggestion we have now repeated the DNA loss experiment with wild type and *mip1^{exo-}* cells compared in parallel. In these new experiments as well, we do not observe nucleoid loss in *mip1^{exo-}* cells under damage (revised Fig. 3C). Furthermore, complementation of this mutant with the Mip1 exonuclease domain results in mtDNA loss, dependent on intact exonuclease activity (revised Fig. 4B-C).

We do acknowledge that there is indeed a modest effect of mtDarT even at 0 hr (pre-induction). This could be due to compromised polymerase activity. However, the extent of mtDNA loss observed upon damage induction is significantly different from the 0 hr time point, and this appears to be dependent on exonuclease activity (see also response to point 3). To ensure a balanced presentation of our observations, in the revised discussion section, we provide a synthesized view of potential mechanisms for mtDNA loss, discuss possible roles for compromised mtDNA replication and mitophagy (see also response to point 2), and highlight clearly the role of Mip1 exonuclease activity in mtDNA clearance under damage (L272-L284).

2. The authors propose that the mtDarT generated adducts trigger exonucleolytic digestion of the mtDNA as evidence by the loss of mtDNA. However, there is no direct experiment presented that demonstrates degradation of the mtDNA through this process. The decline in mtDNA could be done in a mitophagy mediated degradation of other means of clearance of the damaged mtDNA not involving degradation. Experiments are needed to see degradation

products such as following the fate of the mtDNA after induction by the mtDarT (in Wt and Mip1^{exo}) by such methods as PCR, Southern blot, and sequencing.

> We thank the reviewer for the raising the point on the role of mitophagy in the mtDNA clearance process. We did investigate a role for mitophagy in our experiments. We observed significant mtDNA loss under damage in cells deleted for Atg32 (revised Fig. EV3A). Furthermore, we did not observe a significant change in Cox4 levels under damage conditions (revised Fig. EV3B). These observations are also consistent with our previous findings that mitophagy (which is typically observed in much longer timescales) may not play a role under mtDNA damage in yeast (Dua et al., 2022, PMID: 36074064). We now include the mitophagy-related experiments in our revised manuscript (Fig. EV3B and L178-L188) .

Additionally, asymmetric partitioning of mitochondria (or mtDNA) during cell division could also contribute to mtDNA loss. We rule out this mechanism as well, by carrying out experiments in cell cycle-arrested cells. In this case too, we observe Mip1 exonuclease-dependent mtDNA loss in non-dividing cells (Fig. 2C, EV2B and L153-L176).

We also highlight that the microscopy-based analysis we have used in our manuscript is a well-accepted method for measuring mtDNA foci in single-cells and this has now been published by several labs (PMID: 26744405, 33454000, 38365818, 28900194). With additional experiments ruling out the contribution of mitophagy and cell division, as well as new mutational analysis of Mip1, we are confident of the conclusions we present in our study.

We did attempt to carry out qPCR measurements of mtDNA loss under damage conditions. These experiments presented some (very frustrating) challenges with mtDNA isolation from cells treated with mtDarT, despite several attempts and variations in DNA extraction methods. We assume that this is due to fragmentation and loss of mtDNA in these conditions, resulting in unreliable extraction of mtDNA that will allow for a well-controlled experiment measuring mtDNA copy number variations in an ensemble approach. In support of the effect of mtDarT on mtDNA loss, we did observe some reduction in mtDNA copy numbers from qPCR measurements of cells grown for a few generations without inducer (likely due to mtDarT accumulation from leaky expression). We present these data for the reviewer in Reviewer Fig. 1. However, given our inability to conduct the experiment in the presence of damage, we do not include these data in the revised manuscript.

Reviewer Fig. 1

Figure for referees not shown.

3. It is unclear whether the two deletion mutants even produce a stably folded protein. To demonstrate that these two deletions of the Mip1 protein are even functional, the authors need to purify the recombinant protein and demonstrate that they still retain exonuclease activity *in vitro*. I suspect that these proteins are not stable *in vitro* and are being degraded. The Western blot in Fig S4 does not demonstrate that the protein is stable and retains exonuclease activity.

> The C Δ 263 mutant is a well-characterized mutant and has already been shown *in vitro* to have exonuclease activity (PMID: 31445096). So as to confirm that exonuclease activity from the Mip1-C Δ 263 construct is indeed contributing to the observed mtDNA loss, we introduced the exonuclease mutation in the truncation construct. Western blot of both the Mip1-C Δ 263 and exonuclease deficient Mip1-C Δ 263 show similar expression levels (revised Fig. EV4B). The mutant proteins also localized to the mitochondria and colocalized with DAPI-stained mt-nucleoids (revised Fig. EV4A). Importantly, under damage, we no longer observed mtDNA loss occurring in the absence of exonuclease activity from this construct. Based on these results, we conclude that Mip1-C Δ 263 does indeed harbor exonuclease activity. These results are now included in Fig. (4A-B) and presented in results section from L240-L246 of the revised manuscript.

We acknowledge that the Δ 669 mutant has not been characterized *in vitro* previously. After some consideration, we have removed data and reference to the Δ 669 mutant in our revised manuscript, as it does not add substantially to the main conclusions of this section. The biochemical characterizations of the Δ 669 mutant are outside the scope of our current study.

4. The model proposed that when Mip1 encounters a DNA lesion, it switches from replication to degradation can be easily tested *in vitro* with purified proteins and labeled adducted primer template.

> This would indeed be an interesting future direction. The focus of the present work was to track the fate of mtDNA *in vivo* under DNA damage. Given this, we have adopted genetic strategies to perturb various pathways and Mip1, and correspondingly quantify changes to the mtDNA numbers using microscopy-based measurements. Our hypothesis regarding Mip1 exonuclease activity in the model comes from detailed studies on Mip1/PolG and other polymerases *in vitro* wherein concepts such as stimulation of exonuclease activity in response to replication hurdles, synergistic interactions between different Mip1/PolG mutations *in trans*, coupling between exonuclease and polymerase activities of the enzyme etc. have been shown biochemically. These studies include (PMID: 32037587; PMID: 22432028, PMID: 20601675 and PMID: 31445096).

Our proposed model incorporates the observations from these studies and our own to explain loss of entire genome sized copies *in vivo*. The consequence of which is a mechanism to rapidly clear damaged DNA copies by the proofreading function of the polymerase. In addition, we tested for potential factors stimulating the Mip1 exonuclease within the mitochondria (such as dNTP levels). Indeed, new *in vitro* studies (in light of our observations) would be needed to dissect the detailed molecular mechanism of Mip1 exonuclease activity. Such a detailed study requires significant time and effort, and this is outside the scope and focus of our present work. We do plan to pursue this avenue in the future as a dedicated project.

To ensure that our *in vivo* observations remain the central focus of our present work, we have toned-down discussions related to the mechanistic aspects of Mip1 activity (L224-L235, L249-L251 from the original manuscript have been removed).

5. In addition to the use of the two deletion mutants, the authors should consider a point mutation in the polymerase active site that decreases nucleotide insertions without compromising cell survival on glycerol. The literature has several examples of homologous human disease mutations that have been studied in yeast that cause more sensitivity of single stranded DNA to damaging agents.

> We thank the reviewer for this suggestion. Based on literature survey, we identified two polymerase mutants of Mip1 (in the Thumb and Finger-domains respectively), that should affect polymerase activity. However, these mutants have been shown to have high petite frequencies *in vivo* (PMID: 17980715, 20185557), and compromised growth on glycerol. We too attempted to make these mutants, and saw similar phenotypes, which would make any analysis under DNA damaging conditions unreliable. As stated in our response to the first comment of the reviewer, apart from any impact on replication, our data strongly support a major role for exonuclease action in the observed mtDNA loss. Given that mtDNA loss does not occur in a *mip1^{exo-}* mutant (where polymerase activity is still present), replication perturbation alone may not contribute immediately to the observed phenotype.

Nonetheless, to assess the contribution of the pol domain to the exonuclease activity, we expressed these mutants from a low copy replicating vector in the Mip1 exonuclease deficient strain. We found that these mutants were able to complement the Mip1 exonuclease mutant, and mtDNA loss was observed. We include these data in Fig. 4C, 4D and L231-L239 of the revised manuscript.

Referee #3:

The manuscript poses intriguing ideas and it is truly a shame that the manuscript does not fulfill scientific standards: unless I missed critical explanations, fundamental flaws in the experimental setup prevent any support for publication from my side:

(1) WT controls are not present or not run in parallel with mutant strains in figures 3, 4, and 5B-C as well as S3A-B and S5. Thus, it is impossible to compare the data and the effects of the mutations. Empty vector controls by themselves are insufficient to assess the effects of the mutations in comparison to WT cells. Unless the experiments in the mentioned figure are repeated with the inclusion of the required controls run in parallel, these data are not publishable. Given these crucial problems, I will not provide a detailed review for this manuscript.

> Following the reviewer's advice, we have performed all the experiments with wild type mtDarT for direct comparison (revised Fig. 2-5 and Fig. EV2-EV5). We also still include the wild type (empty vector) control data, as additional support for our observations. Our observations are as reported in our original manuscript.

(2) The authors published previously that the expression of a *mip1-exo-* variant would accelerate mtDNA loss upon mtDarT expression compared with wildtype MIP1 expressing cells (Dua et al., JCB 2022). However, these data are not even mentioned or discussed in the current manuscript. Why do the authors observe opposite effects in these experiments?

> We apologise that our writing was not clear in this regard, and thank the reviewer for bringing this important point up. Our earlier work on characterizing mtDarT (Dua et al., 2022) was done in growth conditions with a fermentable carbon source. Hence yeast cells can continue to grow and divide even in the absence of functional mitochondria/ mtDNA. In that study too, we had shown that Mip1 exonuclease activity played a role in mtDNA clearance; in its absence, clearance rates were accelerated via asymmetric mtDNA segregation and cell division, resulting in rapid generation of rho0 cells. Indeed, when cell cycle was arrested in *mip1^{exo-}* cells growing in YPD, we did not observe mtDNA loss (PMID: 36074064, Fig. 5F).

In our current work, we specifically studied the role of Mip1 in mtDNA clearance in a non-fermentable carbon source, where cells would need functional mitochondria/ mtDNA to for proliferation. Importantly, there is no discrepancy between our earlier and present work, with both studies lending strong support to the observation that cells require Mip1 for mtDNA clearance under damage. In case of growth on fermentable carbon source, yeast cells utilize cell division as an additional means to drive this clearance (specifically when Mip1 exonuclease is compromised, Dua et al., 2022). When the possibility of viable cell divisions is compromised (non-fermentable carbon source), Mip1 exonuclease appears to be the sole driver of this clearance during mtDNA stress.

We realize that we should have highlighted this more clearly in our manuscript. We have hence included a section in the discussion synthesizing evidences from both studies (Dua 2022 and present study) to highlight the role of Mip1 and the importance of mtDNA clearance under conditions of mtDNA damage (L315-L323).

Dear Anjana,

Thank you for submitting your revised manuscript. It has now been seen by two of the original referees. Please accept my apologies for this unusual delay. As mentioned before, it took longer than anticipated to receive the referee reports.

As you can see, both referees find that the study is significantly improved during revision and recommend publication. However, I need you to address the points below before I can accept the manuscript.

- Please add a point in Discussion about the possible mechanisms by which mtDNA is degraded (as per referee #2).
- Please address the remaining minor concerns of referee #1.
- Please add a link into the Data Availability, which directly resolves to the S-BIAD1418 dataset.
- Please place the Data Availability section before the Acknowledgements section.
- Please reduce the number of keywords to 5.
- Please rename the Declaration of interests section as Disclosure Statement and Competing Interests.
- Please remove the Author Contributions section from the manuscript.
- As per our format requirements, in the reference list, citations should be listed in alphabetical order and then chronologically, with the authors' surnames and initials inverted; where there are more than 10 authors on a paper, 10 will be listed, followed by 'et al.'. Please see <https://www.embopress.org/page/journal/14693178/authorguide#referencesformat>
- Please fill out and include an author checklist as listed in our online guidelines (<https://www.embopress.org/page/journal/14693178/authorguide>)
- We note that Fig 5C is currently not called out in the text.
- All research articles submitted as revised versions must include a structured methods section that includes a Reagents and Tools Table followed by a Methods and Protocols section. Please see <https://www.embopress.org/page/journal/14693178/authorguide#structuredmethods> for further information.
- Related to the point above, I think Tables EV1-4 are supposed to be parts of the Reagents and Tools table.
- As per formatting criteria, Table S5 is better suited for Dataset file type. The correct nomenclature is Dataset EV1. Please update their source file names, titles in the manuscript tracking system, figure legends in the manuscript, callouts in the manuscript. (Please see <https://www.embopress.org/page/journal/14693178/authorguide#expandedview>).
- The source data folders for EV figures should be grouped into one zip folder.
- Please rename Materials and methods section as Methods.
- Our production/data editors have asked you to clarify several points in the figure legends:
 - o Please note that the exact p values are not provided in the legends of figures 2C, 3C, 4A-E; 5B, EV2 B; EV3 A, C.
 - o Please indicate the statistical test used for data analysis in the legends of figures 2C, 3C, 4A-E; 5B, D; EV2 B; EV3A-D; EV5 A.
 - o Please note that the error bars are not defined in the legends of figures EV3 A, C.
 - o Please note that the scale bar needs to be defined for figures 2A, B; 3B; EV4 A, E
 - o Please note that scale bar and its definition are missing for figures EV1C, EV1 D, E; EV2 A; EV5 B.
 - o Please note that the 2A, B; 3B; EV1 D, E; EV2 A; EV4 A, E dotted lines are not defined in the legend of figure. This needs to be rectified.
- Papers published in EMBO Reports include a 'synopsis' and 'bullet points' to further enhance discoverability. Both are displayed on the html version of the paper and are freely accessible to all readers. The synopsis includes a short standfirst summarizing the study in 1 or 2 sentences (max 35 words) that summarize the paper and are provided by the authors and streamlined by the handling editor. I would therefore ask you to include your synopsis blurb and 3-5 bullet points listing the key experimental findings.
- In addition, please provide an image for the synopsis. This image should provide a rapid overview of the question addressed in the study but still needs to be kept fairly modest since the image size cannot exceed 550 (width) x 300-600 (height) pixels.

Thank you again for giving us to consider your manuscript for EMBO Reports, I look forward to your minor revision.

Kind regards,

Deniz

--

Deniz Senyilmaz Tiebe, PhD
Senior Scientific Editor
EMBO Reports

Referee #1:

I have reviewed this manuscript previously for the EMBO Journal (reviewer #1). The authors have addressed my previous concerns in an adequate manner. Publication of this interesting manuscript can now be recommended.

The following minor points should be corrected before publication:

1. Add a reference for Atg32 in line 182.
2. Lines 180, 306: *Drosophila*, capital D, italics
3. Line 225: MIP1 promoter, capital letters, italics

Referee #2:

In this revision, the authors appear to have addressed most of the previous reviewers comments. However, there are still a few concerns. There is still little mechanistic insight on how the mtDNA gets degraded. See below.

The degradation by the 3'-5' exonuclease activity of Mip1 would require a nick or break in the mtDNA. The authors have not provided the source of this nick or break. Do the authors propose this as an intermediate in BER? The problem is that BER does not function on single strand DNA and since DarT only acts on ssDNA, what is generating the strand break. This is a serious oversight in the paper.

Perhaps the mtDNA digestion and repair represents a specialized repair due to the ribosylation of ssDNA. However, this would require much more experimentation to establish.

All editorial and formatting issues were resolved by the authors.

Dr. Anjana Badrinarayanan
National Centre for Biological Sciences
GKVK Campus
Bellary Road
Bangalore 560065
India

Dear Anjana,

Thank you for submitting your revised manuscript and sending us the revised synopsis image. I have now looked at everything and all is fine. Therefore, I am very pleased to accept your manuscript for publication in EMBO Reports.

Congratulations on a nice work!

Kind regards,

Deniz

--

Deniz Senyilmaz Tiebe, PhD
Senior Scientific Editor
EMBO Reports

--
